# QUANTEASE: OPTIMIZATION-BASED QUANTIZATION FOR LARGE LANGUAGE MODELS

## ABSTRACT

With the rising popularity of Large Language Models (LLMs), there has been an increasing interest in compression techniques that enable their efficient deployment. This study focuses on the Post-Training Quantization (PTQ) of LLMs. Drawing from recent advances, our work introduces `QuantEase`, a layer-wise quantization framework where individual layers undergo separate quantization. The problem is framed as a discrete-structured non-convex optimization, prompting the development of algorithms rooted in Coordinate Descent (CD) techniques. These CD-based methods provide high-quality solutions to the complex non-convex layer-wise quantization problems. Notably, our CD-based approach features straightforward updates, relying solely on matrix and vector operations, circumventing the need for matrix inversion or decomposition. We also explore an outlier-aware variant of our approach, allowing for retaining significant weights (outliers) with complete precision. Our proposal attains state-of-the-art performance regarding perplexity and zero-shot accuracy in empirical evaluations across various LLMs and datasets, with relative improvements of up to 15% over methods such as GPTQ. Particularly noteworthy is our outlier-aware algorithm's capability to achieve near or sub-3-bit quantization of LLMs with an acceptable drop in accuracy, obviating the need for non-uniform quantization or grouping techniques, improving upon methods such as SpQR by up to two times in terms of perplexity.

## 1 INTRODUCTION

Recent years have witnessed an explosive emergence of Large Language Models (LLMs) (Devlin et al., 2018; Radford et al., 2019; Brown et al., 2020; Zhang et al., 2022; Laurençon et al., 2022; Touvron et al., 2023a;b) and their ability to solve complex language modelling tasks for settings like zero-shot or instruction fine-tuning (OpenAI, 2023; Wei et al., 2022a). Consequently, there has been increased interest to utilize LLMs for real-world use cases. The success of LLMs can be attributed to an increase in training data size and the number of model parameters (Kaplan et al., 2020). As a result, modern LLMs have ballooned to hundreds of billions of parameters in size (Brown et al., 2020; Zhang et al., 2022). While the ability of these models to solve tasks is remarkable, efficiently serving them remains a formidable challenge due to their memory footprint. Another notable challenge is increased inference latency, which proves detrimental in practice (Frantar et al., 2023).

Model compression has emerged as a viable approach to tackle the critical challenges of storage footprint and inference speed for LLMs (Hoefler et al., 2021). Within the modern landscape of deep learning research, numerous techniques exist that leverage weight sparsification or quantization to compress large models (Rastegari et al., 2016; Bulat and Tzimiropoulos, 2019; Agustsson et al., 2017; Benbaki et al., 2023). Since modern LLMs take significant compute resources, time, and potentially millions of dollars to train, compression-aware re-training is generally not practicable. This makes post-training quantization (PTQ) an attractive proposition. While numerous practical PTQ algorithms have already been developed (Frantar and Alistarh, 2022; Hubara et al., 2021; Nagel et al., 2020), it is only in the most recent past that algorithms capable of effectively and efficiently quantizing and/or sparsifying extremely large LLMs have become available. Among such methods, prominent techniques include GPTQ (Frantar et al., 2023), SpQR (Dettmers et al., 2023) and AWQ (Lin et al., 2023), among others. These methods aim to compress a 16-bit model into 3 or

4 bits while striving to maintain predictive accuracy. Despite promising progress in this realm, a discernible drop in the performance of quantized models persists compared to their unquantized counterparts. In this paper, we focus on developing a new algorithm for PTQ of LLMs.

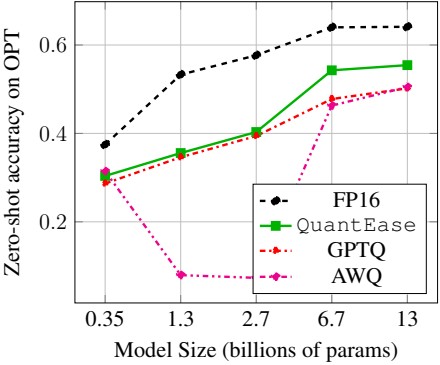 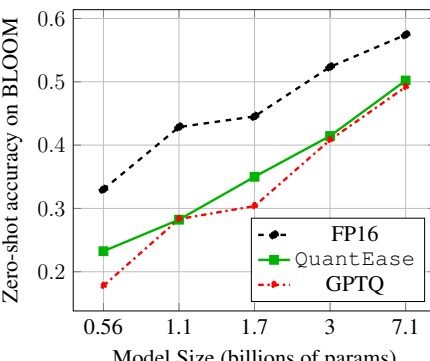

Figure 1: Zero-Shot accuracy on the LAMBADA (Paperno et al., 2016) benchmark for 3-bit quantization. See Section 4 for more details on experimental setup. `QuantEase` consistently outperforms methods like GPTQ and AWQ, sometimes by 15% in terms of relative improvement.

**Our Approach** In this paper, we propose **QuantEase** - a new algorithm to obtain high-quality feasible (i.e. quantized) solutions to layerwise post-training quantization of LLMs. `QuantEase` leverages cyclic Coordinate Descent (CD) (Tseng, 2001), which has traditionally been used to address statistical problems (Chang and Lin, 2011; Mazumder and Hastie, 2012; Friedman et al., 2010; Hazimeh and Mazumder, 2020; Shevade and Keerthi, 2003; Behdin et al., 2023). We show that `QuantEase` is a more principled optimization method when compared to methods like GPTQ, guaranteeing a non-increasing sequence of the objective value under feasiblity of the initial solution. `QuantEase` cycles through coordinates in each layer, updating each weight such that it minimizes the objective while keeping other weights fixed. This allows for simple closed-form updates for each weight. Empirically, this results in up to 30% improvement in relative quantization error over GPTQ for 4-bit and 3-bit quantization (see Figure 2).

`QuantEase`'s simplicity allows it to have several advantages over other methods, while making it extremely scalable and easy to use. Unlike other methods, `QuantEase` does not require expensive matrix inversion or Cholesky factorization. This helps avoid numerical issues while also lowering memory requirements. `QuantEase` is extremely easy to implement, making it a drop-in replacement for most other methods. Remarkably, `QuantEase` can efficiently quantize a 66B parameter model on a single Nvidia V100 GPU, whereas methods like GPTQ and AWQ run out of memory. Our experiments also reveal that `QuantEase` can quantize extremely large models with tens of billions of parameters in a matter of several hours to up to a day, making it a viable algorithm for practical use.

`QuantEase`'s effectiveness in achieving lower quantization error also translates to improvements on language modeling benchmarks. Experiments on several LLM model families (Laurençon et al., 2022; Zhang et al., 2022) and language tasks show that `QuantEase` outperforms state-of-the-art uniform quantization methods such as GPTQ and AWQ in terms of perplexity and zero-shot accuracy (Paperno et al., 2016), both in the 4-bit and 3-bit regimes (see Tables 1, 2 and Figures 1, 3). For the 3-bit regime, `QuantEase` is especially effective for zero-shot accuracy, achieving strong relative improvements (up to 15%) over GPTQ (see Figure 1).

We also propose a variant of `QuantEase` to handle weight outliers. This is achieved by dividing the set of layer weights into a group of quantized weights and very few unquantized ones. We propose a block coordinate descent method based on iterative hard thresholding (Blumensath and Davies, 2009) for the outlier-aware version of our method. This version of `QuantEase` is able to improve upon outlier-based methods such as SpQR. Particularly, we show this outlier-aware method can achieve acceptable accuracy for sub-3 bit regimes, improving upon current available methods such as SpQR by up to 2.5 times in terms of perplexity. We hope that the simplicity and effectiveness of `QuantEase` inspires further research in this area.

Our contributions can be summarised as follows:

***Optimization-based PTQ*** - We propose `QuantEase`, an optimization framework for post-training quantization of LLMs based on minimizing a layerwise (least squares) reconstruction error. We propose a CD framework updating network weights one-at-a-time, avoiding memory-expensive matrix inversions/factorizations. Particularly, we make the CD updates efficient by exploiting the problem structure resulting in closed form updates for weights. Notably, `QuantEase` can quantize models with up to 66B parameters on a single V100 GPU.

***Outlier awareness*** - We also propose an outlier-aware version of our framework where a few (outlier) weights are kept unquantized. We discuss an algorithm for outlier-aware `QuantEase` based on block coordinate descent and iterative hard thresholding.

***Improved accuracy*** - Experiments on LLMs with billions of parameters show that `QuantEase` outperforms recent PTQ methods such as GPTQ and AWQ in text generation and zero-shot tasks for 3 and 4-bit quantization, often by a large margin. Additional experiments show that the outlier-aware `QuantEase` outperforms methods such as SpQR by up to 2.5 times in terms of perplexity, in the near-3 or sub-3 bit quantization regimes.

**Related Work**    Recently, there has been a mounting interest in the layerwise quantization of LLMs. One prominent method for post-training quantization of LLMs is GPTQ (Frantar et al., 2023). GPTQ extends the Optimal Brain Surgeon (OBS) framework (LeCun et al., 1989; Hassibi and Stork, 1992; Frantar and Alistarh, 2022), incorporating strategies for layerwise compression (Dong et al., 2017). Empirical evaluations of GPTQ demonstrate encouraging results, revealing a marginal drop in accuracy in text generation and zero-shot benchmarks. Another avenue for achieving layer-wise quantization of LLMs is the recent work by Lin et al. (2023), referred to as AWQ. This approach centres on preserving the weights that influence activations most. GPTQ and AWQ represent two prominent foundational techniques, and we will compare our methodology against these in our experiments. We present a detailed review of earlier work in Appendix A.

It is widely acknowledged that transformer models, inclusive of LLMs, confront challenges tied to outliers when undergoing quantization to lower bit-widths (Wei et al., 2022b; Bondarenko et al., 2021; Kim et al., 2023). This predicament arises from the notable impact of extremely large or small weights on the quantization range, thereby leading to supplementary errors. As a result, specific research endeavours delve into the notion of non-uniform quantization. SqueezeLLM (Kim et al., 2023) seeks to identify and preserve outlier weights (for example, very large or small weights) that might affect the output the most, allowing for improved accuracy. Similarly, SpQR (Dettmers et al., 2023) combines GPTQ with outlier detection to achieve a lower loss of accuracy. We use SpQR as a benchmark in our experiments with outlier detection.

A long line of work discuss and study quantization methods for large neural networks from an implementation perspective, including hardware-level optimizations for low-bit calculations and inference time activation quantization, which are beyond the scope of this paper. For a more exhaustive exposition, please check Gholami et al. (2022); Wang et al. (2020); Hubara et al. (2021); Xiao et al. (2023); Yao et al. (2022; 2023) and the references therein.

**Problem Formulation: Layerwise Quantization**    A standard approach to LLM compression is layerwise compression (Dong et al., 2017), where layers are compressed/quantized one at a time. This allows the task of compressing a very large network to be broken down into compressing several smaller layers, which is more practical than simultaneously quantizing multiple layers. In this paper, we pursue a layerwise framework for quantization.

Focusing on layerwise quantization, let us consider a linear layer with some (nonlinear) activation function. Within this context, let $\boldsymbol{X} \in \mathbb{R}^{p \times n}$ represent the input matrix feeding into this particular layer, where $p$ denotes the number of input features and $n$ denotes the number of training data points fed through the network. Additionally, let $\boldsymbol{W} \in \mathbb{R}^{q \times p}$ symbolize the weights matrix corresponding to this layer, characterized by $q$ output channels.

For a given output channel $i \in [q]$, we designate the predetermined, finite set of per-channel quantization levels for this channel as $\mathcal{Q}_i \subseteq \mathbb{R}$. In this work, following Frantar et al. (2023), we focus on the case where the quantization levels within $\mathcal{Q}_i$ are uniformly spaced (We note however, that our approach can extend to quantization schemes that do not follow this assumption.). We can then

formulate the layerwise quantization task as the following optimization problem:

$$\min_{\hat{\boldsymbol{W}}} f(\hat{\boldsymbol{W}}) := \|\boldsymbol{W}\boldsymbol{X} - \hat{\boldsymbol{W}}\boldsymbol{X}\|_F^2 \text{ s.t. } \hat{W}_{i,j} \in \mathcal{Q}_i, (i,j) \in [q] \times [p]. \tag{1}$$

The objective of Problem (1) captures the distance between the original pre-activation output obtained from unquantized weights ($\boldsymbol{W}\boldsymbol{X}$), and the pre-activation output stemming from quantized weights ($\hat{\boldsymbol{W}}\boldsymbol{X}$), subject to $\hat{\boldsymbol{W}}$ adhering to quantization constraints.

**Remark 1.** We emphasize that the channel-wise uniform quantization setting that we study here is the same as the setting in, for example, GPTQ (Frantar et al., 2023). This makes our method a direct replacement for, say, GPTQ.

**Notation:** For $i \in [q] := \{1, \ldots, q\}$, we define the quantization operator $q_i$ with respect to quantization levels $\mathcal{Q}_i$ as follows:

$$q_i(x) \in \underset{y \in \mathcal{Q}_i}{\operatorname{argmin}} \ (x - y)^2. \tag{2}$$

For a matrix such as $\boldsymbol{A}$, $\|\boldsymbol{A}\|_F$ denotes its Frobenius norm. Moreover, for $i < j$, $\boldsymbol{A}_{:,i}$ and $\boldsymbol{A}_{:,i:j}$ denote the $i$-th column, and columns $i$ to $j$ of $\boldsymbol{A}$, respectively.

## 2 OUR PROPOSED METHOD

Our algorithm, `QuantEase`, is based on the cyclic CD method (Tseng, 2001). At every update of our CD algorithm, we minimize the objective in (1) with respect to the coordinate $\hat{W}_{i,j}$ while making sure $\hat{W}_{i,j} \in \mathcal{Q}_i$ (we keep all other weights fixed at their current value). Mathematically, for $(i,j) \in [q] \times [p]$ we update $\hat{W}_{i,j}$ as follows:

$$\hat{W}_{i,j}^+ \in \underset{\hat{W}_{i,j} \in \mathcal{Q}_i}{\operatorname{argmin}} f(\hat{W}_{1,1}, \cdots, \hat{W}_{i,j}, \cdots, \hat{W}_{q,p}) \tag{3}$$

where $\hat{\boldsymbol{W}}^+$ is the solution after the update of coordinate $(i, j)$. In words, $\hat{W}_{i,j}^+$ is obtained by solving a 1D optimization problem: we minimize the 1D function $\hat{W}_{i,j} \mapsto f(\hat{\boldsymbol{W}})$ under the quantization constraint. This 1D optimization problem, despite being non-convex, can be solved to optimality in closed-form (See Lemma 1 for details). A full pass over all coordinates $(i,j) \in [q] \times [p]$ completes one iteration of the CD algorithm. `QuantEase` usually makes several iterations to obtain a good solution to (1).

We note that regardless of the initialization used for CD, after one iteration of CD we obtain a feasible (i.e. quantized) solution. Moreover, from the second iteration onward, feasibility is maintained by `QuantEase`, while continuously decreasing $f$. This is useful because `QuantEase` can be terminated any time after the first iteration with a feasible solution.

**Closed-form updates:** The efficiency of the CD method depends on how fast the update (3) can be calculated. Lemma 1 derives a closed form solution for Problem (3).

**Lemma 1.** Let $\boldsymbol{\Sigma} = \boldsymbol{X}\boldsymbol{X}^T$. Then, $\hat{W}_{i,j}^+ = q_i(\tilde{\beta})$ in (3) where[1]

$$\tilde{\beta} = - \left[ \sum_{k \neq j} \Sigma_{j,k} \hat{W}_{i,k} - (\boldsymbol{W}\boldsymbol{\Sigma})_{i,j} \right] / \Sigma_{j,j}. \tag{4}$$

Proof of Lemma 1 is found in Appendix C. We note that $\tilde{\beta}$ in (4) minimizes the one-dimensional function $\hat{W}_{i,j} \mapsto f(\hat{W}_{1,1}, \cdots, \hat{W}_{i,j}, \cdots, \hat{W}_{q,p})$ where $\hat{W}_{i,j}$ is unconstrained (i.e., without any quantization constraint). Thus, Lemma 1 shows that to find the best quantized value for $\hat{W}_{i,j}$ in (3), it suffices to quantize the value that minimizes the one-dimensional function, $\hat{W}_{i,j} \mapsto f(\hat{W}_{1,1}, \cdots, \hat{W}_{i,j}, \cdots, \hat{W}_{q,p})$ under no quantization constraint. As we find the minimizer per coordinate and then quantize the minimizer, this is different from quantizing the "current" weight.

---

[1]We assume that $\Sigma_{j,j} > 0$. Note that $\Sigma_{j,j} = 0$ would mean that $\boldsymbol{X}_{j,:} = \boldsymbol{0}$; hence, $\hat{\boldsymbol{W}}_{:,j}$ may be quantized arbitrarily and completely omitted from the problem. Such checks can be done before `QuantEase` is begun.

**Parallelization over** $i \in [q]$**:** As seen in Lemma 1, for a given $j_0 \in [p]$, the updates of $\hat{W}_{i,j_0}$ are independent (for each $i$) and can be done simultaneously. Therefore, rather than updating a coordinate $\hat{W}_{i,j}$ at a time, we update a column of $\hat{W}$, that is: $\hat{W}_{:,j}$ at each update. This allows us to better make use of the problem structure (see the rank-1 update below).

**Rank-1 updates:** Note that in (4), we need access to terms of the form $\sum_{k \neq j} \Sigma_{j,k} \hat{W}_{i,k}$. Such terms can be refactored as:

$$\sum_{k \neq j} \Sigma_{j,k} \hat{W}_{i,k} = \sum_{k=1}^{p} \Sigma_{j,k} \hat{W}_{i,k} - \Sigma_{j,j} \hat{W}_{i,j} = \hat{W}_{i,:} \mathbf{\Sigma}_{:,j} - \Sigma_{j,j} \hat{W}_{i,j}.$$

However, as noted above, we update all the rows corresponding to a given column of $\hat{W}$ at once. Therefore, to update a column of $\hat{W}$, we need access to the vector

$$\left( \sum_{k \neq j} \Sigma_{j,k} \hat{W}_{1,k}, \cdots, \sum_{k \neq j} \Sigma_{j,k} \hat{W}_{q,k} \right)^T = (\hat{W}\mathbf{\Sigma})_{:,j} - \Sigma_{j,j} \hat{W}_{:,j}. \tag{5}$$

Drawing from (5), maintaining a record of $\hat{W}\mathbf{\Sigma}$ exclusively for the updates outlined in (4) emerges as satisfactory, given that $W$ and $\mathbf{\Sigma}$ remain unaltered in the iterative process demonstrated in (4). Below, we show how the updates of $\hat{W}\mathbf{\Sigma}$ can be done with a notable degree of efficiency. First, consider the following observation.

*Observation:* Suppose $W_1, W_2$ differ only on a single column such as $j$. Then,

$$W_2 \mathbf{\Sigma} = \underbrace{[W_1 \mathbf{\Sigma} - (W_1)_{:,j} \mathbf{\Sigma}_{j,:}]}_{(A)} \underbrace{+ (W_2)_{:,j} \mathbf{\Sigma}_{j,:}}_{(B)}. \tag{6}$$

Thus, given $W_1 \mathbf{\Sigma}$, obtaining $W_2 \mathbf{\Sigma}$ requires two rank-1 updates, rather than a full matrix multiplication. We apply these updates to keep track of $\hat{W}\mathbf{\Sigma}$ when updating a column of $\hat{W}$, as shown in Algorithm 1. Additional implementation details of `QuantEase` (including our custom initialization) are discussed in Appendix D.1

---

**Algorithm 1:** `QuantEase`

---

Initialize $\hat{W}$
**for** *iter* $= 1, \cdots,$ *iter-max* **do**
    **for** $j = 1, \cdots, p$ **do**
        $\boldsymbol{u} \leftarrow \left[ (\hat{W}\mathbf{\Sigma})_{:,j} - \Sigma_{j,j} \hat{W}_{:,j} - (W\mathbf{\Sigma})_{:,j} \right] / \Sigma_{j,j}$ // $\tilde{\beta}$ from Lemma 1 for
            column $j$
        $\hat{W}\mathbf{\Sigma} \leftarrow \hat{W}\mathbf{\Sigma} - \hat{W}_{:,j} \mathbf{\Sigma}_{j,:}$ // Part $(A)$ of rank-1 update from (6)
        $\hat{W}_{i,j} \leftarrow q_i(-u_i), i \in [q]$ // Perform updates from (4)
        $\hat{W}\mathbf{\Sigma} \leftarrow \hat{W}\mathbf{\Sigma} + \hat{W}_{:,j} \mathbf{\Sigma}_{j,:}$ // Part $(B)$ of rank-1 update from (6)
    **end**
**end**
**return** $\hat{W}$

---

### 2.1 CONVERGENCE OF QUANTEASE

Next, we discuss the convergence of `QuantEase`. Let us define Coordinate-Wise (CW) minima.

**Definition 1** (CW-minimum, Beck and Eldar (2013); Hazimeh and Mazumder (2020))**.** *We call $W^*$ a CW-minimum for Problem* (1) *iff for* $(i,j) \in [q] \times [p]$*, we have* $W_{i,j}^* \in \mathcal{Q}_i$ *and*

$$W_{i,j}^* \in \underset{\hat{W}_{i,j} \in \mathcal{Q}_i}{\operatorname{argmin}} f(\hat{W}_{1,1}, \cdots, \hat{W}_{i,j}, \cdots, \hat{W}_{q,p}).$$

In words, a CW-minimum is a feasible solution that cannot be improved by updating only one coordinate of the solution, while keeping the rest fixed. Suppose we modify the basic CD update, (3) as follows: *If $\hat{W}_{i,j}^{+}$ does not strictly decrease $f$, then set $\hat{W}_{i,j}^{+} = \hat{W}_{i,j}$.* This avoids oscillations of the algorithm with a fixed $f$ value. The following lemma shows that the sequence of weights from the modified CD converges to a CW-minimum.

**Lemma 2.** *The sequence of $\hat{W}$ generated from modified* `QuantEase` *converges to a CW-minimum.*

## 2.2 Optimization performance: GPTQ vs QuantEase

We now show that `QuantEase` indeed leads to lower (calibration) optimization error compared to GPTQ (see Section 4 for the experimental setup details). To this end, for a given layer and a feasible solution $\hat{W}$, let us define the relative calibration error as $\text{Error}(\hat{W}) = \|WX - \hat{W}X\|_F^2 / \|WX\|_F^2$ where $X$ is the calibration set used for quantization.

In Figure 2, we report the relative error of `QuantEase`, $\text{Error}(\hat{W}^{\text{QuantEase}})$, as well as the relative improvement of `QuantEase` over GPTQ in terms of error, $(\text{Error}(\hat{W}^{\text{GPTQ}}) - \text{Error}(\hat{W}^{\text{QuantEase}}))/\text{Error}(\hat{W}^{\text{GPTQ}})$ for the BLOOM-1b1 model and 3/4 bit quantization. In the figure, we sort layers based on their `QuantEase` error, from the smallest to the largest. As can be seen, the `QuantEase` error over different layers can differ between almost zero to 5% for the 4-bit and zero to 15% for the 3-bit quantization. This shows different layers can have different levels of compressibility. Moreover, we see that `QuantEase` in almost all cases improves upon GPTQ, achieving a lower optimization error (up to 30%). This shows the benefit of `QuantEase`.

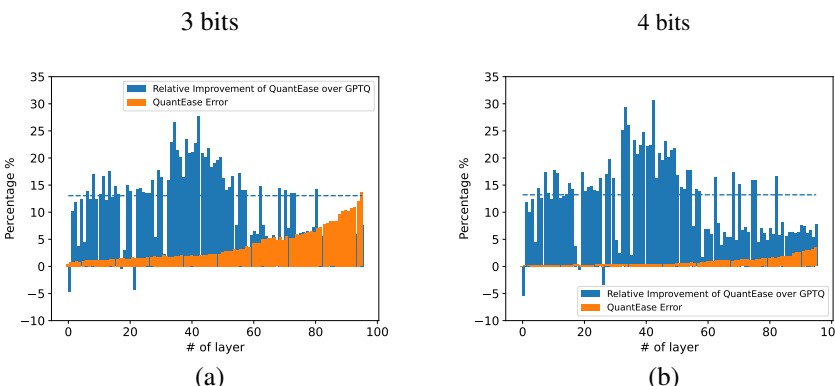

Figure 2: Comparison of the optimization performance of `QuantEase` and GPTQ over all layers. The horizontal dashed line shows the median improvement of `QuantEase` over GPTQ for each case. As can be seen, `QuantEase` results in lower optimization error compared to GPTQ for most layers (up to 30% and on median 12%). Moreover, we see that the error in 3 bit quantization is larger than 4 bit quantization.

## 3 Outlier-Aware Quantization

**Formulation** It has been observed that the activation might be more sensitive to some weights in a layer over others (Dettmers et al., 2023; Yao et al., 2022). Finding a suitable value on the quantization grid might not be possible for such sensitive weights, leading to substantial quantization errors. Moreover, some weights of a pre-trained network can be significantly larger or smaller than the rest—the quantization of LLMs can be affected by such weights (Wei et al., 2022b; Bondarenko et al., 2021; Kim et al., 2023). The existence of large/small weights increases the range of values that need to be quantized, which in turn increases quantization error. To this end, to better handle sensitive and large/small weights, which we collectively call outlier weights, we first introduce a modified version of the layerwise quantization problem (1). Our optimization formulation identifies a collection of weights kept in complete precision (aka the outliers weights) and quantizes the remaining weights.

Before presenting our outlier-aware quantization formulation, we introduce some notation. Let $\mathcal{S} \subseteq [q] \times [p]$ denote a set of outlier indices—the corresponding weights are left at full precision. For any $(i, j) \notin \mathcal{S}$, the $(i, j)$-th weight is quantized and is chosen from the quantization grid $\mathcal{Q}_i$ for the $i$-th channel. This is equivalent to substituting the set of weights for the layer $\boldsymbol{W}$ with $\hat{\boldsymbol{W}} + \hat{\boldsymbol{H}}$ where $\hat{\boldsymbol{W}}$ is quantized (like in Problem (1)) and $\hat{\boldsymbol{H}}$ is sparse, with only a few nonzeros. In particular, for any $(i, j) \notin \mathcal{S}$, we have $\hat{H}_{i,j} = 0$ implying the $(i, j)$-th weight can only have a quantized component. As $\mathcal{S}$ has a small cardinality, $\hat{\boldsymbol{H}}$ is mostly zero. On the other hand, when $(i, j) \in \mathcal{S}$, we have $\hat{H}_{i,j} \neq 0$ and the corresponding weight is retained at full precision. In light of the above discussion, we present an outlier-aware version of Problem (1) given by:

$$\min_{\hat{\boldsymbol{W}}, \hat{\boldsymbol{H}}} g(\hat{\boldsymbol{W}}, \hat{\boldsymbol{H}}) := \|\boldsymbol{W}\boldsymbol{X} - (\hat{\boldsymbol{W}} + \hat{\boldsymbol{H}})\boldsymbol{X}\|_F^2 \text{ s.t. } \hat{W}_{i,j} \in \mathcal{Q}_i \ (i,j) \in [q] \times [p], \quad \|\hat{\boldsymbol{H}}\|_0 \leq s \quad (7)$$

where $\|\cdot\|_0$ denotes the number of nonzero elements of a vector/matrix, and $s \ll p, q$ is the total budget on the number of outliers. The constraint $\|\hat{\boldsymbol{H}}\|_0 \leq s$ ensures the total number of outliers remains within the specified limit.

**Remark 2.** We note that the setup of dividing weights into to a set of quantized weights and a few outliers is standard and similar to the setting studied by SpQR (Dettmers et al., 2023), making our method a drop-in replacement for SpQR.

**Optimizing Problem (7)** To obtain good solutions to Problem (7), we use a block coordinate descent method where we alternate between updating $\hat{\boldsymbol{W}}$ (with $\hat{\boldsymbol{H}}$ fixed) and then $\hat{\boldsymbol{H}}$ (with $\hat{\boldsymbol{W}}$ fixed). For a fixed $\hat{\boldsymbol{H}}$, Problem (7) has the same form as $f(\hat{\boldsymbol{W}})$ in (1) where $\boldsymbol{W}\boldsymbol{X}$ is substituted with $(\boldsymbol{W} - \hat{\boldsymbol{H}})\boldsymbol{X}$. Therefore, we can use `QuantEase` as discussed in Section 2 to update $\hat{\boldsymbol{W}}$. Next, we discuss how to update $\hat{\boldsymbol{H}}$. For a fixed $\hat{\boldsymbol{W}}$, Problem (7) is a least squares problem with a cardinality constraint. We use proximal gradient method (aka iterative hard thresholding method) (Blumensath and Davies, 2009) where we make a series of updates of the form:

$$\hat{\boldsymbol{H}}^+ \in \operatorname*{argmin}_{\boldsymbol{K} \in \mathbb{R}^{q \times p}} \tilde{g}(\boldsymbol{K}) \text{ s.t. } \|\boldsymbol{K}\|_0 \leq s = P_s(\hat{\boldsymbol{H}} - \eta \nabla_{\boldsymbol{H}} g(\hat{\boldsymbol{W}}, \hat{\boldsymbol{H}})) \quad (8)$$

where $P_s(\boldsymbol{A})$ sets all coordinates of $\boldsymbol{A}$ to zero except the $s$-largest in absolute value,

$$\tilde{g}(\boldsymbol{K}) = \frac{L}{2} \left\| \boldsymbol{K} - \left( \hat{\boldsymbol{H}} - \frac{1}{L} \nabla_{\hat{\boldsymbol{H}}} g(\hat{\boldsymbol{W}}, \hat{\boldsymbol{H}}) \right) \right\|_F^2, \quad (9)$$

$L = 1/\eta = 2\lambda_{\max}(\boldsymbol{X}\boldsymbol{X}^T)$ and $\lambda_{\max}(\boldsymbol{A})$ is the largest eigenvalue of the matrix $\boldsymbol{A}$. Lemma 3 below establishes updates in (8) form a descent method if the initial $\hat{\boldsymbol{H}}$ is sparse, $\|\hat{\boldsymbol{H}}\|_0 \leq s$.

**Lemma 3.** *For any $\hat{\boldsymbol{W}}$ and any $\hat{\boldsymbol{H}}$ such that $\|\hat{\boldsymbol{H}}\|_0 \leq s$, we have $g(\hat{\boldsymbol{W}}, \hat{\boldsymbol{H}}^+) \leq g(\hat{\boldsymbol{W}}, \hat{\boldsymbol{H}})$.*

We note that unlike SpQR which fixes the location of outliers after selecting them, our method is able to add new outlier coordinates or remove them as the optimization progresses. This is because the location of nonzeros of $\hat{\boldsymbol{H}}$ (i.e. outliers) gets updated, in addition to their values. Additional implementation details of the outlier-aware method is discussed in Appendix D.2, with the summary of algorithm given in Algorithm 2.

## 4 EXPERIMENTS

In this section, we conduct several numerical experiments to demonstrate the effectiveness of `QuantEase`. A PyTorch implementation of `QuantEase` will be released in the near future.

**Setup** We follow an experimental setup mostly similar to the one in Frantar et al. (2023). We use 128 sequences from C4 data (Raffel et al., 2020) as our training (calibration) data, and consider several models from OPT (Zhang et al., 2022) and BLOOM (Laurençon et al., 2022) families. For uniform quantization, we compare `QuantEase` to RTN (Yao et al., 2022; Dettmers et al., 2022), GPTQ (Frantar et al., 2023) and AWQ (Lin et al., 2023). For methods related to outlier detection, we compare with SpQR (Dettmers et al., 2023). In our experiments, we do not use grouping for

Table 1: OPT family perplexity for WikiText2 quantized on C4. OOM indicates running out of memory for GPTQ and AWQ. `QuantEase` achieves lower perplexity in the majority of settings.

|  |  | 350m | 1.3b | 2.7b | 6.7b | 13b | 66b |
|---|---|---|---|---|---|---|---|
| full |  | 22.00 | 14.62 | 12.47 | 10.86 | 10.13 | 9.34 |
| 3 bits | RTN | 64.56 | 1.33e4 | 1.56e4 | 6.00e3 | 3.36e3 | 6.12$e$3 |
|  | AWQ | $32.38_{0.11}$ | $53.63_{0.45}$ | $201_{6}$ | $19.00_{0.12}$ | $13.90_{0.02}$ | OOM |
|  | GPTQ | $33.60_{0.34}$ | $21.51_{0.13}$ | $17.02_{0.17}$ | $15.16_{0.01}$ | $\mathbf{11.90}_{0.06}$ | OOM |
|  | QuantEase | $\mathbf{31.52}_{0.36}$ | $\mathbf{21.30}_{0.23}$ | $\mathbf{16.75}_{0.24}$ | $\mathbf{12.95}_{0.04}$ | $12.41_{0.02}$ | $\mathbf{13.08}_{0.38}$ |
| 4 bits | RTN | 25.94 | 48.19 | 16.92 | 12.10 | 11.32 | 110.52 |
|  | AWQ | $24.05_{0.03}$ | $15.67_{0.04}$ | $13.16_{0.01}$ | $11.30_{0.01}$ | $10.36_{0.01}$ | OOM |
|  | GPTQ | $24.29_{0.11}$ | $15.44_{0.03}$ | $\mathbf{12.80}_{0.04}$ | $11.46_{0.04}$ | $10.34_{0.01}$ | OOM |
|  | QuantEase | $\mathbf{23.91}_{0.05}$ | $\mathbf{15.28}_{0.04}$ | $13.05_{0.01}$ | $\mathbf{11.21}_{0.01}$ | $\mathbf{10.32}_{0.01}$ | $\mathbf{9.47}_{0.02}$ |

Table 2: BLOOM family perplexity for WikiText2 quantized on C4. `QuantEase` achieves lower perplexity in the majority of settings.

|  |  | 560m | 1b1 | 1b7 | 3b | 7b1 |
|---|---|---|---|---|---|---|
| full |  | 22.41 | 17.68 | 15.39 | 13.48 | 11.37 |
| 3 bits | RTN | 56.99 | 50.07 | 63.50 | 39.29 | 17.37 |
|  | GPTQ | $32.36_{0.07}$ | $25.18_{0.06}$ | $21.43_{0.07}$ | $17.50_{0.04}$ | $13.73_{0.03}$ |
|  | QuantEase | $\mathbf{31.52}_{0.10}$ | $\mathbf{23.91}_{0.02}$ | $\mathbf{20.03}_{0.05}$ | $\mathbf{17.21}_{0.04}$ | $\mathbf{13.43}_{0.04}$ |
| 4 bits | RTN | 25.89 | 19.98 | 16.97 | 14.75 | 12.10 |
|  | GPTQ | $24.02_{0.03}$ | $\mathbf{18.90}_{0.02}$ | $16.41_{0.02}$ | $\mathbf{14.10}_{0.01}$ | $11.74_{0.01}$ |
|  | QuantEase | $\mathbf{23.97}_{0.03}$ | $\mathbf{18.90}_{0.01}$ | $\mathbf{16.11}_{0.03}$ | $14.18_{0.01}$ | $\mathbf{11.69}_{0.01}$ |

any method as our focus is on understanding the optimization performance of various methods. Moreover, as discussed by Kim et al. (2023); Yao et al. (2023), grouping can lead to additional inference-time overhead, reducing the desirability of such tricks in practice. Our experiments were conducted on a single NVIDIA V100 GPU with 32GB of memory.

**Additional Experiments:** Appendix B contains additional numerical results related to the effect of number of iterations of `QuantEase`, runtime and text generation.

**Language Generation Benchmarks**   We study the effect of quantization on language generation tasks. Perplexity results evaluated on WikiText2 data (Merity et al., 2016) and OPT/BLOOM.[2] families are shown in Tables 1 and 2, respectively. Perplexity results evaluated on PTB data (Marcus et al., 1994) can be found in Tables B.1 and B.2 in the appendix. `QuantEase` achieves lower perplexity in all cases, except for OPT-13b, for 3-bit quantization. In the 4-bit regime, `QuantEase` almost always either improves upon baselines or achieves similar performance. Since perplexity is a stringent measure of model quality, these results demonstrate that `QuantEase` results in better quantization compared to methods like GPTQ and AWQ.

**LAMBADA Zero-shot Benchmark**   Following (Frantar et al., 2023), we compare the performance of our method with baselines over a zero-shot task, namely LAMBADA (Paperno et al., 2016). The results for this task are shown in Figure 3 for the OPT and BLOOM families. For 3-bit quantization, `QuantEase` outperforms GPTQ and AWQ, often by a large margin. In the 4-bit regime, the performance of all methods is similar, although `QuantEase` seems to be overall the most performant.

**Outliers-Aware Performance**   Next, we study the performance of the outlier-aware version of `QuantEase`. To this end, we consider 3-bit quantization and two sparsity levels of 0.5% and 1%

---

[2]We do not implement AWQ for BLOOM due to known architectural issues. See `https://github.com/mit-han-lab/llm-awq/issues/2` for more details.

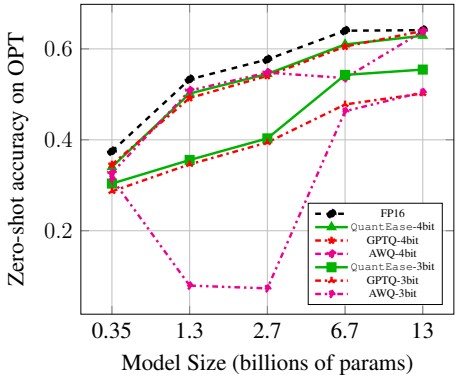 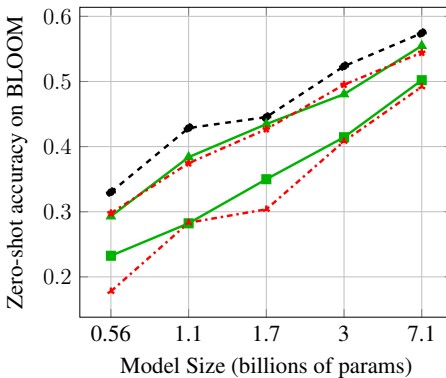

Figure 3: Zero-Shot accuracy on the LAMBADA (Paperno et al., 2016) benchmark for 3-bit and 4-bit quantization. See Section 4 for more details on experimental setup.

Table 3: OPT family perplexity for WikiText2 quantized on C4. Outlier aware quantization is done with 3 bits and 2 bits. QuantEase beats SpQR, often by a large margin.

|  |  | 350m | 1.3b | 2.7b | 6.7b | 13b |
|---|---|---|---|---|---|---|
| full |  | 22.00 | 14.62 | 12.47 | 10.86 | 10.13 |
| 3 bits | QuantEase | $31.52_{0.12}$ | $21.30_{0.23}$ | $16.75_{0.24}$ | $12.95_{0.04}$ | $12.41_{0.02}$ |
| Outlier (3 bits) | SpQR 1% | $31.67_{0.43}$ | $18.17_{0.16}$ | $14.50_{0.07}$ | $11.95_{0.02}$ | $10.96_{0.01}$ |
|  | QuantEase 0.5% | $27.52_{0.05}$ | $16.68_{0.14}$ | $13.72_{0.04}$ | $11.49_{0.02}$ | $10.70_{0.01}$ |
|  | QuantEase 1% | $\mathbf{26.48}_{0.12}$ | $\mathbf{16.25}_{0.05}$ | $\mathbf{13.70}_{0.10}$ | $\mathbf{11.48}_{0.03}$ | $\mathbf{10.37}_{0.01}$ |
| Outlier (2 bits) | SpQR 2% | $323_{7}$ | $155_{3}$ | $70.5_{2.7}$ | $34.0_{1.2}$ | $22.3_{0.2}$ |
|  | QuantEase 2% | $\mathbf{158}_{4}$ | $\mathbf{36.4}_{0.8}$ | $\mathbf{24.2}_{0.1}$ | $\mathbf{19.0}_{0.2}$ | $\mathbf{19.3}_{0.2}$ |

(for example, $s = 0.005pq$ or $s = 0.01pq$). Roughly speaking, a $0.5\%$ outlier budget would lead to an additional 0.15 bits overhead (i.e. 3.15 bits on average), while the $1\%$ version would lead to an additional overhead of 0.3 bits (i.e. 3.3 bits on average). We compare our method with SpQR, with the threshold tuned to have at least $1\%$ outliers on average. The rest of the experimental setup is shared from previous experiments. The perplexity results (on WikiText2) for this comparison are reported in Table 3 for the OPT family and in Table B.3 for the BLOOM family. As is evident from the results, the QuantEase $0.5\%$ outlier version is able to significantly outperform SpQR in all cases, and the $1\%$ method does even better. This shows that outlier-aware QuantEase makes near-3 bit quantization possible without the need for any grouping.

Next, we study extreme quantization of models to the 2-bit regime. Particularly, we consider the base number of bits of 2 and $2\%$ outliers, resulting in roughly 2.6 bits on average. The results for this experiment for OPT are shown in Table 3 and for BLOOM in Table B.4. QuantEase significantly outperforms SpQR and is able to maintain acceptable accuracy in the sub-3-bit quantization regime. We observed empirically that in the absence of grouping and outlier detection, if we were to do 2-bit quantization, then the resulting solutions lead to a significant loss of accuracy. This finding also appears to be consistent with our exploration of other methods' publicly available code, such as GPTQ and AWQ.

## 5 FUTURE WORK

In this work, we did not consider grouping. However, grouping can be easily incorporated in QuantEase, as we only use standard quantizers. Investigating the performance of QuantEase with grouping is left for a future study. Moreover, we note that QuantEase can be paired with AWQ. As Lin et al. (2023) note, incorporating AWQ into GPTQ can lead to improved numerical results, and as we have shown, QuantEase usually outperforms GPTQ. Therefore, we would expect AWQ+QuantEase would lead to even further improvements.

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

# A   DETAILED LITERATURE REVIEW

## A.1   GPTQ

As mentioned earlier, GPTQ (Frantar et al., 2023) extends the OBS-based framework of Frantar and Alistarh (2022); Hassibi and Stork (1992). GPTQ performs quantization of $\boldsymbol{W}$ one column at a time. Specifically, GPTQ starts with the initialization, $\hat{\boldsymbol{W}} \leftarrow \boldsymbol{W}$. Then, it cycles through columns $j = 1, \cdots, p$ and for each $j$, it quantizes column $j$ of $\hat{\boldsymbol{W}}$. For the $j$-th column it quantizes all its entries via the updates: $\hat{W}_{i,j}^+ = q_i(\hat{W}_{i,j})$, $i \in [q]$. After updating the $j$-th column, GPTQ proceeds to update the other weights in the layer to ensure that the error in (1) does not increase too much. To make our exposition resemble that of the OBS framework, we note that GPTQ updates $\hat{\boldsymbol{W}}_{:,j+1:p}$ by approximately solving the least-squares problem:

$$\min_{\hat{\boldsymbol{W}}_{:,j:p}} \|\boldsymbol{W}\boldsymbol{X} - \hat{\boldsymbol{W}}\boldsymbol{X}\|_F^2 \quad \text{s.t.} \quad \hat{\boldsymbol{W}}_{:,j} = \hat{\boldsymbol{W}}_{:,j}^+, \tag{A.1}$$

where the constraint above implies that we are effectively optimizing over $\hat{\boldsymbol{W}}_{:,j+1:p}$ (but we choose this representation following Hassibi and Stork (1992); Frantar and Alistarh (2022)). We note that in (A.1), the quantization constraints are dropped as otherwise, (A.1) would be as hard as (1) in general. Moreover, entries up to the $j$-th column i.e, $\hat{\boldsymbol{W}}_{:,1:j}$ are not updated to ensure they remained quantized. Since the OBS framework is set up to optimize a homogeneous quadratic[3], Frantar and Alistarh (2022); Frantar et al. (2023) reformulate Problem (A.1) as

$$\min_{\hat{\boldsymbol{W}}_{:,j:p}} \|\boldsymbol{A} + (\boldsymbol{W}_{:,j} - \hat{\boldsymbol{W}}_{:,j})\boldsymbol{X}_{j,:} + (\boldsymbol{W}_{:,j+1:p} - \hat{\boldsymbol{W}}_{:,j+1:p})\boldsymbol{X}_{j+1:p,:}\|_F^2 \quad \text{s.t.} \quad \hat{\boldsymbol{W}}_{:,j} = \hat{\boldsymbol{W}}_{:,j}^+ \tag{A.2}$$

where

$$\boldsymbol{A} = (\boldsymbol{W}_{:,1:j-1} - \hat{\boldsymbol{W}}_{:,1:j-1})\boldsymbol{X}_{1:j-1,:}. \tag{A.3}$$

Upon inspection one can see that the objective in (A.2) is not homogeneous quadratic, and hence does not fit into the OBS framework. Therefore, $\boldsymbol{A}$ is dropped and this problem is replaced by the formulation:

$$\min_{\hat{\boldsymbol{W}}_{:,j:p}} \|(\boldsymbol{W}_{:,j} - \hat{\boldsymbol{W}}_{:,j})\boldsymbol{X}_{j,:} + (\boldsymbol{W}_{:,j+1:p} - \hat{\boldsymbol{W}}_{:,j+1:p})\boldsymbol{X}_{j+1:p,:}\|_F^2 \quad \text{s.t.} \quad \hat{\boldsymbol{W}}_{:,j} = \hat{\boldsymbol{W}}_{:,j}^+$$

$$\stackrel{(a)}{=} \min_{\hat{\boldsymbol{W}}_{:,j:p}} \mathsf{Tr}((\boldsymbol{W}_{:,j:p} - \hat{\boldsymbol{W}}_{:,j:p})^T \boldsymbol{\Sigma}_F (\boldsymbol{W}_{:,j:p} - \hat{\boldsymbol{W}}_{:,j:p})) \quad \text{s.t.} \quad \hat{\boldsymbol{W}}_{:,j} = \hat{\boldsymbol{W}}_{:,j}^+ \tag{A.4}$$

where $\boldsymbol{\Sigma}_F = \boldsymbol{X}_{j:p,:}\boldsymbol{X}_{j:p,:}^T$, $F$ refers to the $\{j, \cdots, p\}$ indices and $(a)$ is by $\|\boldsymbol{M}\|_F^2 = \mathsf{Tr}(\boldsymbol{M}^T\boldsymbol{M})$ for any matrix $\boldsymbol{M}$. Therefore, after updating the $j$-th column to $\hat{\boldsymbol{W}}_{:,j}^+$, we can update $\hat{\boldsymbol{W}}_{:,j+1:p}$ by the OBS updates (we refer to Frantar et al. (2023); Frantar and Alistarh (2022) for derivation details):

$$\boldsymbol{\delta} \leftarrow -\frac{\hat{\boldsymbol{W}}_{:,j} - \hat{\boldsymbol{W}}_{:,j}^+}{[\boldsymbol{\Sigma}_F^{-1}]_{j,j}}$$

$$\hat{\boldsymbol{W}}_{:,j+1:p} \leftarrow \hat{\boldsymbol{W}}_{:,j+1:p} + \boldsymbol{\delta}[\boldsymbol{\Sigma}_F^{-1}]_{j,j+1:p} \tag{A.5}$$

We note that the OBS updates in (A.5) require the calculation of $\boldsymbol{\Sigma}_F^{-1}$. Therefore, using the updates in (A.5) can be expensive in practice. To improve efficiency, GPTQ uses a lazy-batch update scheme where at each step, only a subset (of size at most 128) of the remaining unquantized weights is updated.

## A.2   AWQ

Similar to GPTQ, AWQ (Lin et al., 2023) uses a layerwise quantization framework. However, different from GPTQ, the main idea behind AWQ is to find a rescaling of weights that does not result

---

[3]A homogeneous quadratic function with decision variable $\boldsymbol{u} \in \mathbb{R}^p$ is given by $\boldsymbol{u}^T \boldsymbol{Q} \boldsymbol{u}$ where $\boldsymbol{Q} \in \mathbb{R}^{p \times p}$ and there is no linear term.

in high quantization error, rather than directly minimizing the least squares criteria for layerwise reconstruction. To this end, AWQ considers the following optimization problem:

$$\min_{\boldsymbol{s}\in\mathbb{R}^p} \|\boldsymbol{W}\boldsymbol{X} - \boldsymbol{q}(\boldsymbol{s}\odot\boldsymbol{W})(\boldsymbol{X}\odot\boldsymbol{s}^{-1})\|_F^2 \tag{A.6}$$

where $[\boldsymbol{q}(\boldsymbol{W})]_{i,j} = q_i(W_{i,j})$ quantizes a vector/matrix coordinate-wise, $[\boldsymbol{s}^{-1}]_i = s_i^{-1}$ is the coordinate-wise inversion and $\odot$ is the channel-wise multiplication, $[\boldsymbol{s}\odot\boldsymbol{W}]_{i,j} = s_j W_{i,j}$ and $[\boldsymbol{X}\odot\boldsymbol{s}^{-1}]_{i,j} = X_{i,j}/s_i$. In Problem (A.6), $\boldsymbol{s}$ is the per-channel scaling. Problem (A.6) is non-differentiable and non-convex and cannot be efficiently solved. Therefore, Lin et al. (2023) discuss grid search heuristics for $\boldsymbol{s}$ to find a value that does not result in high quantization error. Particularly, they set $\boldsymbol{s} = \boldsymbol{s_X}^\alpha * \boldsymbol{s_W}^{-\beta}$ for some $\alpha, \beta \in [0,1]$, where $*$ is coordinate-wise multiplication, and $\boldsymbol{s_X}, \boldsymbol{s_W} \in \mathbb{R}^p$ are per-channel averages of magnitude of $\boldsymbol{X}$ and $\boldsymbol{W}$, respectively. The values of $\alpha, \beta$ are then chosen by grid search over the interval $[0,1]$. After choosing the value of $\boldsymbol{s}$, the quantized weights are given as $\boldsymbol{s}^{-1} \odot \boldsymbol{q}(\boldsymbol{s}\odot\boldsymbol{W})$.

### A.3 SPQR

Fianlly, we review SpQR (Dettmers et al., 2023) which incorporates sensitivity-based quantization into GPTQ. Particularly, they seek to select few outliers that result in higher quantization error and keep them in full-precision. To this end, for each coordinate $(i,j) \in [q] \times [p]$ SpQR calculates the sensitivity to this coordinate as the optimization error resulting from quantizing this coordinate. Formally, they define sensitivity as

$$\omega_{ij} = \min_{\hat{\boldsymbol{W}}} \|\boldsymbol{W}\boldsymbol{X} - \hat{\boldsymbol{W}}\boldsymbol{X}\|_F^2 \quad \text{s.t.} \quad \hat{W}_{i,j} = q_i(W_{i,j}). \tag{A.7}$$

We note that Problem (A.7) is in OBS form and therefore OBS is then used to calculate the sensitivity of coordinate $(i,j)$. Then, any coordinate that has high sensitivity, for example, $\omega_{i,j} > \tau$ where $\tau > 0$ is a predetermined threshold, is considered to be an outlier. After selecting outliers, similar to GPTQ, SpQR cycles through columns $j = 1, \cdots, p$ and updates each column based on OBS updates (see Section A.1), keeping outlier weights in full-precision.

## B    NUMERICAL RESULTS

### B.1    EFFECT OF NUMBER OF ITERATIONS

We first study the effect of the number of iterations of `QuantEase` on model performance. To this end, we consider OPT-350m in 3/4 bits and run quantization for different numbers of iterations, ranging from 10 to 30. The perplexity on WikiText2 is shown in Figure B.1 for this case. The results show that increasing the number of iterations generally lowers perplexity as `QuantEase` reduces the error, although the improvement in perplexity for 4-bit quantization is small. Based on these results, 25 iterations seem to strike a good balance between accuracy and runtime, which we use in the rest of our experiments.

Effect of number of iterations

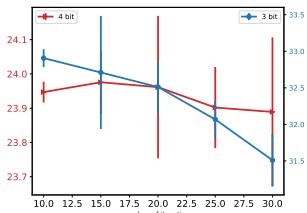

Figure B.1: Effect of varying the number of iterations of `QuantEase` on perplexity in Section B.1.

## B.2 Additional Text Generation Benchmarks

First, we present the results for uniform quantization evaluated on PTB dataset (the setup from Section 4). These results can be found in Tables B.1 and B.2. Additional outlier aware quantization results are presented in Tables B.3 and B.4 for BLOOM family from Section 4.

| | | 350m | 1.3b | 2.7b | 6.7b | 13b | 66b |
|---|---|---|---|---|---|---|---|
| full | | 26.08 | 16.96 | 15.11 | 13.09 | 12.34 | 11.36 |
| 3 bits | RTN | 81.09 | 1.16e4 | 9.39e3 | 4.39e3 | 2.47e3 | 3.65e3 |
| | AWQ | $40.20_{0.07}$ | $98.12_{2.26}$ | $188_4$ | $26.07_{0.14}$ | $19.96_{0.05}$ | OOM |
| | GPTQ | $39.28_{0.04}$ | $26.36_{0.14}$ | $19.98_{0.13}$ | $18.86_{0.02}$ | $\mathbf{13.88}_{0.03}$ | OOM |
| | QuantEase | $\mathbf{37.70}_{0.15}$ | $\mathbf{25.24}_{0.27}$ | $\mathbf{19.90}_{0.05}$ | $\mathbf{15.78}_{0.06}$ | $13.89_{0.02}$ | $\mathbf{12.93}_{0.19}$ |
| 4 bits | RTN | 31.12 | 34.15 | 22.11 | 16.09 | 15.39 | 274.56 |
| | AWQ | $29.30_{0.02}$ | $18.82_{0.08}$ | $16.40_{0.01}$ | $13.86_{0.01}$ | $12.77_{0.01}$ | OOM |
| | GPTQ | $28.84_{0.09}$ | $18.44_{0.01}$ | $\mathbf{15.87}_{0.01}$ | $13.78_{0.05}$ | $12.59_{0.01}$ | OOM |
| | QuantEase | $\mathbf{28.49}_{0.10}$ | $\mathbf{18.23}_{0.04}$ | $15.95_{0.01}$ | $\mathbf{13.56}_{0.03}$ | $\mathbf{12.50}_{0.01}$ | $\mathbf{11.55}_{0.01}$ |

Table B.1: OPT family perplexity for PTB quantized on C4

| | | 560m | 1b1 | 1b7 | 3b | 7b1 |
|---|---|---|---|---|---|---|
| full | | 41.23 | 46.96 | 27.92 | 23.12 | 19.40 |
| 3 bits | RTN | 117.17 | 151.76 | 115.10 | 59.87 | 32.03 |
| | GPTQ | $64.63_{0.65}$ | $72.57_{0.70}$ | $42.48_{0.04}$ | $31.36_{0.1}$ | $24.34_{0.05}$ |
| | QuantEase | $\mathbf{59.37}_{0.40}$ | $\mathbf{69.61}_{0.51}$ | $\mathbf{38.92}_{0.30}$ | $\mathbf{30.59}_{0.2}$ | $\mathbf{23.73}_{0.06}$ |
| 4 bits | RTN | 48.56 | 54.51 | 31.20 | 25.39 | 20.92 |
| | GPTQ | $44.35_{0.09}$ | $51.64_{0.07}$ | $30.10_{0.06}$ | $24.33_{0.01}$ | $20.21_{0.04}$ |
| | QuantEase | $\mathbf{43.90}_{0.04}$ | $\mathbf{51.60}_{0.34}$ | $\mathbf{29.50}_{0.17}$ | $\mathbf{24.22}_{0.03}$ | $\mathbf{20.11}_{0.01}$ |

Table B.2: BLOOM family perplexity for PTB quantized on C4

| | | 560m | 1b1 | 1b7 | 3b | 7b1 |
|---|---|---|---|---|---|---|
| full | | 22.41 | 17.68 | 15.39 | 13.48 | 11.37 |
| 3 bits | QuantEase | $31.52_{0.10}$ | $23.91_{0.02}$ | $20.03_{0.05}$ | $17.21_{0.04}$ | $13.43_{0.04}$ |
| Outlier (3 bits) | SpQR 1% | $29.02_{0.05}$ | $22.51_{0.11}$ | $19.16_{0.05}$ | $15.95_{0.06}$ | $12.88_{0.01}$ |
| | QuantEase 0.5% | $26.82_{0.07}$ | $20.26_{0.03}$ | $17.41_{0.02}$ | $14.93_{0.01}$ | $12.19_{0.01}$ |
| | QuantEase 1% | $\mathbf{25.80}_{0.09}$ | $\mathbf{19.60}_{0.02}$ | $\mathbf{17.06}_{0.02}$ | $\mathbf{14.65}_{0.02}$ | $\mathbf{12.03}_{0.01}$ |
| 4 bits | QuantEase | $23.97_{0.03}$ | $18.90_{0.01}$ | $16.11_{0.03}$ | $14.18_{0.01}$ | $11.69_{0.01}$ |

Table B.3: BLOOM family perplexity for WikiText2 quantized on C4. Outlier aware quantization is done with 3 bits.

## B.3 Generative Examples

In this section, we present a few generative examples. We have used OPT-13b model quantized to 4 bits in all examples here.

### B.3.1 Example 1

- **Prompt:** `Every time data scientist Kim ran her new algorithm, it kept outputting the same unexpected result: a recipe for`
- **FP16:** `Every time data scientist Kim ran her new algorithm, it kept outputting the same unexpected result: a recipe for a new`

| | | 560m | 1b1 | 1b7 | 3b | 7b1 |
|---|---|---|---|---|---|---|
| full | | 22.41 | 17.68 | 15.39 | 13.48 | 11.37 |
| Outlier (2 bits) | SpQR 2% | $228_{10}$ | $126_5$ | $127_4$ | $59.7_{2.2}$ | $32.5_{0.4}$ |
| | QuantEase 2% | $\mathbf{66.1}_{1.3}$ | $\mathbf{39.3}_{0.3}$ | $\mathbf{31.4}_{0.1}$ | $\mathbf{22.1}_{0.1}$ | $\mathbf{15.8}_{0.03}$ |

Table B.4: BLOOM family perplexity for WikiText2 quantized on C4. Outlier aware quantization is done with 2 bits.

```
  kind of chocolate cake. The algorithm was designed to predict
  the likelihood that a person would be a repeat customer of a
  particular restaurant. But it kept coming up with a recipe for a
  chocolate cake.  "I was like, 'What'?" Kim said. "I was like,
  'This is not what I asked for'".
```

- GPTQ: ```Every time data scientist Kim ran her new algorithm, it
  kept outputting the same unexpected result: a recipe for a new
  kind of chocolate cake. Kim, who asked to be identified by her
  first name only, was working on a project to predict the
  likelihood of a person being a criminal. She was using a machine
  learning algorithm to analyze data from the FBI's National Crime
  Information Center, which contains information on more than
  10 million criminal records.```

- QuantEase: ```Every time data scientist Kim ran her new algorithm, it
  kept outputting the same unexpected result: a recipe for a new
  drug. The algorithm was designed to predict the likelihood that
  a drug would be approved by the US Food and Drug
  Administration (FDA). But it kept coming up with a recipe for
  a new drug that was never approved. "It was a bit of a shock,"
  says Kim, a data scientist at the University of California,
  San Francisco.```

In this case, although GPTQ's response is initially similar to the FP16 response (both outputting chocolate cake), as the response continues, GPTQ and FP16 diverge. On the other hand, QuantEase produces a response that is overall coherent and logical. This is while GPTQ's response relates food to crime.

### B.3.2 EXAMPLE 2

- Prompt: ```In a parallel universe where cats are the rulers of Earth,
  our feline king was pondering his first decree```

- FP16: ```In a parallel universe where cats are the rulers of Earth,
  our feline king was pondering his first decree. "I decree that
  all cats shall be given the right to vote," he said. "And I
  decree that all cats shall be given the right to vote," said
  the cat next to him.```

- GPTQ: ```In a parallel universe where cats are the rulers of Earth,
  our feline king was pondering his first decree."I decree that
  all cats shall be allowed to roam free," he said. "And I decree
  that all cats shall be allowed to eat as much as they want,"
  said a second cat. "And I decree that all cats shall be allowed
  to sleep as much as they want," said a third cat.```

- QuantEase: ```In a parallel universe where cats are the rulers of Earth,
  our feline king was pondering his first decree. "I decree
  that all cats shall be allowed to sleep on the bed," he said.
  "But only if they are on their backs," said his advisor. "But
   only if they are on their backs," said the king.```

In this case, both GPTQ and `QuantEase` diverge from the FP16 model, although the response from both seems to be coherent. However, we note that `QuantEase` is able to mention the "advisor" and "king" keeping the context relevant, while GPTQ refers to "second cat" and so on.

### B.3.3 EXAMPLE 3

- **Prompt:** `Data scientist Grace was working late one night when she discovered a glitch in the matrix of her data. It spelled out`

- **FP16:** `Data scientist Grace was working late one night when she discovered a glitch in the matrix of her data. It spelled out the word "HELP" in the form of a series of numbers. She was able to use the glitch to access a hidden database that contained the names and addresses of all the people who had ever used the service.`

- **GPTQ:** `Data scientist Grace was working late one night when she discovered a glitch in the matrix of her data. It spelled out the word "HELP" in the form of a series of numbers. She was so excited that she called her friend, a programmer, and told him about it. "I'm not sure what you're talking about," he said. "I've never seen that before." "Well,`

- **QuantEase:** `Data scientist Grace was working late one night when she discovered a glitch in the matrix of her data. It spelled out the word "HELP" in the form of a series of numbers. She was able to use the glitch to access a hidden database that contained the names of all the people who had ever been in the database. She was able to access the names of all the people who had ever been in the database.`

In this case, `QuantEase` seems to be closer to the FP16 response, although `QuantEase` repeats itself at the end of the sentence, showing `QuantEase` might lose the attention in this case.

### B.3.4 EXAMPLE 4

- **Prompt:** `Last Tuesday, just as the sun was setting, my neighbor's dog, Rover, started to speak to me in perfect French. He said`

- **FP16:** `Last Tuesday, just as the sun was setting, my neighbor's dog, Rover, started to speak to me in perfect French. He said, "I'm sorry, I don' speak English." I was shocked. I've lived in this neighborhood for over a year and Rover has never spoken to me before. I've never seen him before. I've never heard him before.`

- **GPTQ:** `Last Tuesday, just as the sun was setting, my neighbor's dog, Rover, started to speak to me in perfect French. He said, "I'm Rover. I'm a dog. I'm a dog." I was stunned. I was speechless. I was in awe. I've been a dog lover all my life. I've always had dogs. I've always loved dogs. I've always wanted to be a`

- **QuantEase:** `Last Tuesday, just as the sun was setting, my neighbor's dog, Rover, started to speak to me in perfect French. He said, "I'm sorry, I'm so sorry." I was in the middle of a conversation with my friend, and I stopped to listen to Rover. He was standing in the middle of the street, looking up at the sky. "I'm sorry, I'm so sorry," he said again.`

In this case, none of the models appear to follow FP16 response.

|  | 350m | 1.3b | 2.7b | 6.7b | 13b |
|---|---|---|---|---|---|
| QuantEase | 26m | 1.41h | 3.63h | 11.47h | 33.8h |

Table B.5: QuantEase runtime for OPT family

|  | 560m | 1b1 | 1b7 | 3b | 7b1 |
|---|---|---|---|---|---|
| QuantEase | 19.7m | 48.7m | 1.66h | 4.13h | 14.3h |

Table B.6: QuantEase runtime for BLOOM family

### B.4 RUNTIME

In this section, we report the runtime of our QuantEase method. The numbers reported are for 3-bit quantization experiments from Tables 1 and 2 for OPT and BLOOM families, respectively. The runtime for different models are reported in Tables B.5 and B.6. We see that the runtime ranges from 10s of minutes for sub-billion models, up to around a day for 13b model. This shows that overall, QuantEase is computationally feasible, specially for models with 10b or fewer parameters.

## C PROOF OF MAIN RESULTS

### C.1 PROOF OF LEMMA 1

Write

$$
\begin{aligned}
f(\hat{\boldsymbol{W}}) &= \|\boldsymbol{W}\boldsymbol{X} - \hat{\boldsymbol{W}}\boldsymbol{X}\|_F^2 \\
&= \left\| \sum_{j=1}^{p} \hat{\boldsymbol{W}}_{:,j}\boldsymbol{X}_{j,:} - \boldsymbol{W}\boldsymbol{X} \right\|_F^2 \\
&\overset{(a)}{=} \sum_{j,k=1}^{p} \mathsf{Tr}(\boldsymbol{X}_{j,:}^T\hat{\boldsymbol{W}}_{:,j}^T\hat{\boldsymbol{W}}_{:,k}\boldsymbol{X}_{k,:}) + \mathsf{Tr}(\boldsymbol{X}^T\boldsymbol{W}^T\boldsymbol{W}\boldsymbol{X}) - 2\sum_{j=1}^{p}\mathsf{Tr}(\boldsymbol{X}_{j,:}^T\hat{\boldsymbol{W}}_{:,j}^T\boldsymbol{W}\boldsymbol{X}) \\
&= \underbrace{\sum_{j,k=1}^{p}(\boldsymbol{X}_{k,:}\boldsymbol{X}_{j,:}^T\hat{\boldsymbol{W}}_{:,j}^T\hat{\boldsymbol{W}}_{:,k})}_{(A)} + \mathsf{Tr}(\boldsymbol{X}\boldsymbol{X}^T\boldsymbol{W}^T\boldsymbol{W}) - 2\underbrace{\sum_{j=1}^{p}\mathsf{Tr}(\boldsymbol{W}\boldsymbol{X}\boldsymbol{X}_{j,:}^T\hat{\boldsymbol{W}}_{:,j}^T)}_{(B)} \quad \text{(C.1)}
\end{aligned}
$$

where $(a)$ is by $\|\boldsymbol{A}\|_F^2 = \mathsf{Tr}(\boldsymbol{A}^T\boldsymbol{A})$ and $(b)$ is by $\mathsf{Tr}(\boldsymbol{A}\boldsymbol{B}) = \mathsf{Tr}(\boldsymbol{B}\boldsymbol{A})$. Next, let us only consider terms in (C.1) that depend on $\boldsymbol{W}_{:,j_0}$ for a given $j_0$. Letting $\boldsymbol{\Sigma} = \boldsymbol{X}\boldsymbol{X}^T$, such terms can be written as

$$
\overbrace{(\boldsymbol{X}_{j_0,:}\boldsymbol{X}_{j_0,:}^T\hat{\boldsymbol{W}}_{:,j_0}^T\hat{\boldsymbol{W}}_{:,j_0})}^{\text{from } (A),\ j=k=j_0} + 2\overbrace{\sum_{k\neq j_0}(\boldsymbol{X}_{j_0,:}\boldsymbol{X}_{k,:}^T\hat{\boldsymbol{W}}_{:,k}^T\hat{\boldsymbol{W}}_{:,j_0})}^{\text{from } (A),\ j \text{ or } k = j_0} - 2\overbrace{\mathsf{Tr}(\boldsymbol{W}\boldsymbol{X}\boldsymbol{X}_{j_0,:}^T\hat{\boldsymbol{W}}_{:,j_0}^T)}^{\text{from } (B),\ j=j_0}
$$

$$
= \sum_{i=1}^{q}\Sigma_{j_0,j_0}\hat{W}_{i,j_0}^2 + 2\sum_{i=1}^{q}\sum_{k\neq j_0}\Sigma_{j_0,k}\hat{W}_{i,j_0}\hat{W}_{i,k} - 2\sum_{i=1}^{q}(\boldsymbol{W}\boldsymbol{\Sigma})_{i,j_0}\hat{W}_{i,j_0}
$$

$$
= \sum_{i=1}^{q}\left\{ \Sigma_{j_0,j_0}\hat{W}_{i,j_0}^2 + 2\sum_{k\neq j_0}\Sigma_{j_0,k}\hat{W}_{i,j_0}\hat{W}_{i,k} - 2(\boldsymbol{W}\boldsymbol{\Sigma})_{i,j_0}\hat{W}_{i,j_0} \right\}. \quad \text{(C.2)}
$$

Therefore, to find the optimal value of $\hat{W}_{i,j}^+$ in (3) for $(i, j_0)$ we need to solve problems of the form

$$
\min_{u\in\mathcal{Q}_i} \Sigma_{j_0,j_0}u^2 + 2\sum_{k\neq j_0}\Sigma_{j_0,k}\hat{W}_{i,k}u - 2(\boldsymbol{W}\boldsymbol{\Sigma})_{i,j_0}u. \quad \text{(C.3)}
$$

**Claim:** If $a > 0$, then

$$\min_{u \in \mathcal{Q}_i} au^2 + bu = q_i(-b/2a).$$

**Proof of Claim:** Write

$$au^2 + bu = a(u + (b/2a))^2 - b^2/(4a)$$

therefore,

$$
\begin{aligned}
\operatorname*{argmin}_{u \in \mathcal{Q}_i} au^2 + bu &= \operatorname*{argmin}_{u \in \mathcal{Q}_i} (u + b/(2a))^2 \\
&= \operatorname*{argmin}_{u \in \mathcal{Q}_i} (u + b/(2a))^2 \\
&= \operatorname*{argmin}_{y \in \mathcal{Q}_i + b/(2a)} y^2 - b/(2a) \\
&= \tilde{q}_i(0) - b/(2a) \\
&= q(-b/(2a))
\end{aligned}
\tag{C.4}
$$

where $\tilde{q}_i$ is the quantization function for the quantization grid $\mathcal{Q}_i + b/(2a) = \{a + b/(2a) : a \in \mathcal{Q}_i\}$. This completes the proof of the claim and the lemma.

## C.2 PROOF OF LEMMA 2

The proof is a result of the observations that (a) the modified algorithm generates a sequence of $\hat{\boldsymbol{W}}$ iterates with decreasing $f$ values (after obtaining the first feasible solution, possibly after the first iteration) and (b) there are only a finite number of choices for $\hat{\boldsymbol{W}}$ on the quantization grid.

## C.3 PROOF OF LEMMA 3

First, note that mapping $\hat{\boldsymbol{H}} \mapsto \nabla_{\boldsymbol{H}} g(\hat{\boldsymbol{W}}, \hat{\boldsymbol{H}})$ is $L$-Lipschitz,

$$\left\| \nabla_{\boldsymbol{H}} g(\hat{\boldsymbol{W}}, \hat{\boldsymbol{H}}_1) - \nabla_{\boldsymbol{H}} g(\hat{\boldsymbol{W}}, \hat{\boldsymbol{H}}_2) \right\|_F \leq L \|\hat{\boldsymbol{H}}_1 - \hat{\boldsymbol{H}}_2\|_F.$$

Therefore, by Lemma 2.1 of Beck and Teboulle (2009) for $\hat{\boldsymbol{W}}, \hat{\boldsymbol{H}}_1, \hat{\boldsymbol{H}}_2$ we have

$$g(\hat{\boldsymbol{W}}, \hat{\boldsymbol{H}}_1) - g(\hat{\boldsymbol{W}}, \hat{\boldsymbol{H}}_2) \leq \frac{L}{2} \left\| \hat{\boldsymbol{H}}_1 - \left( \hat{\boldsymbol{H}}_2 - \frac{1}{L} \nabla_{\hat{\boldsymbol{H}}} g(\hat{\boldsymbol{W}}, \hat{\boldsymbol{H}}_2) \right) \right\|_F^2 - \frac{1}{2L} \left\| \nabla_{\hat{\boldsymbol{H}}} g(\hat{\boldsymbol{W}}, \hat{\boldsymbol{H}}_2) \right\|_F^2. \tag{C.5}$$

Particularly, from the definition of $\tilde{g}$ in (9), we have

$$g(\hat{\boldsymbol{W}}, \boldsymbol{K}) - g(\hat{\boldsymbol{W}}, \hat{\boldsymbol{H}}) \leq \tilde{g}(\boldsymbol{K}) - \tilde{g}(\hat{\boldsymbol{H}}).$$

By setting $\boldsymbol{K} = \hat{\boldsymbol{H}}^+$ we get

$$g(\hat{\boldsymbol{W}}, \hat{\boldsymbol{H}}^+) - g(\hat{\boldsymbol{W}}, \hat{\boldsymbol{H}}) \leq \tilde{g}(\hat{\boldsymbol{H}}^+) - \tilde{g}(\hat{\boldsymbol{H}}) \leq 0 \tag{C.6}$$

where the second inequality is by the definition of $\hat{\boldsymbol{H}}^+$ in (8) as $\hat{\boldsymbol{H}}$ is a feasible solution for the optimization problem in (8).

# D ADDITIONAL IMPLEMENTATION DETAILS

## D.1 DETAILS OF QUANTEASE

**Initialization:** We initialize QuantEase with original unquantized weights. However, we include the following heuristic in QuantEase. In every other third iteration, we do not quantize weights (i.e. use $\tilde{\beta}$ from Lemma 1 directly). Though it introduces infeasibility, the following iteration brings back feasibility. We have observed that this heuristic helps with optimization performance, i.e., decreases $f$ better.

**Memory Footprint:** The matrices, $\boldsymbol{\Sigma}$ and $\boldsymbol{W}\boldsymbol{\Sigma}$ do not change over iterations and can be stored with $p^2 + \mathcal{O}(pq)$ memory footprint. This is specially interesting as in practice, $n \gg p, q$. QuantEase,

unlike GPTQ, also does not require matrix inversion or Cholesky factorization which can be memory-inefficient (either adding up to $\mathcal{O}(p^2)$ storage). In our experiment for very large models, matrix inversion and Cholesky factorization can lead to out-of-memory issues for GPTQ.

**Computational Complexity:** In each iteration of `QuantEase` for a fixed $j$, the time complexity is dominated by rank-1 updates and is $\mathcal{O}(pq)$. Therefore, each iteration of `QuantEase` has time complexity of $\mathcal{O}(p^2q)$. Combined with the initial cost of computing $\boldsymbol{\Sigma} = \boldsymbol{XX}^T$, $\boldsymbol{W\Sigma}$, $\hat{\boldsymbol{W}}\boldsymbol{\Sigma}$ and doing $K$ iterations, the overall time complexity of `QuantEase` is $\mathcal{O}(pqn + Kp^2q)$.

### D.2 DETAILS OF OUTLIER-AWARE `QUANTEASE`

We note that when calculating $\mathcal{Q}_i$'s for Problem (7), we remove the top $s$ largest coordinates of $\boldsymbol{W}$ (in absolute value) from the quantization pool, as the effect of those weights can be captured by $\hat{\boldsymbol{H}}$ and we do not need to quantize them. This allows to reduce the range that each $\mathcal{Q}_i$ needs to quantize, leading to lower error. Therefore, simultaneously, we preserve sensitive weights and reduce the quantization range by using outlier-aware `QuantEase`.

**Initialization:** In terms of initialization, similar to basic `QuantEase`, we set $\hat{\boldsymbol{H}}, \hat{\boldsymbol{W}}$ such that $\hat{\boldsymbol{H}} + \hat{\boldsymbol{W}} = \boldsymbol{W}$. Particularly, we use the $s$-largest coordinates of $\boldsymbol{W}$ (in absolute value) to initialize $\hat{\boldsymbol{H}}$: $\hat{\boldsymbol{H}} = P_s(\boldsymbol{W})$, $\hat{\boldsymbol{W}} = \boldsymbol{W} - \hat{\boldsymbol{H}}$. Note that this leads to an infeasible initialization of $\hat{\boldsymbol{W}}$ similar to basic `QuantEase`. However, as discussed, after one iteration of `QuantEase` the solution becomes feasible and the descent property of Algorithm 2 holds.

**Memory and Computational Complexity:** We also note that as seen from Algorithm 2, in addition to storing $\hat{\boldsymbol{H}}$, we need to store $\hat{\boldsymbol{H}}\boldsymbol{\Sigma}$, showing the memory footprint remains $p^2 + \mathcal{O}(pq)$, like basic `QuantEase`. In terms of computational complexity, in addition to basic `QuantEase`, the outlier-aware version requires calculating the largest eigenvalue of $\boldsymbol{XX}^T$, which can be done by iterative power method in $\mathcal{O}(p^2)$ only using matrix/vector multiplication. Additionally, calculating $\hat{\boldsymbol{H}}\boldsymbol{\Sigma}$ requires $\mathcal{O}(p^2q)$ in each iteration, and finding the largest $s$ coordinates of $\hat{\boldsymbol{H}}$ can be done with average complexity of $\mathcal{O}(pq \log pq)$. Therefore, the overall complexity is $\mathcal{O}(pqn + Kp^2q + Kpq \log pq)$ for $K$ iterations.

---

**Algorithm 2:** Outlier-Aware `QuantEase`

---

Initialize $\hat{\boldsymbol{H}}, \hat{\boldsymbol{W}}$
$\eta \leftarrow 1/2\lambda_{\max}(\boldsymbol{XX}^T)$ // step size for iterative thresholding
**for** *iter* $= 1, \cdots,$ *iter-max* **do**
  **for** $j = 1, \cdots, p$ **do**
    $\boldsymbol{u} \leftarrow \left[ (\hat{\boldsymbol{W}}\boldsymbol{\Sigma})_{:,j} - \Sigma_{j,j}\hat{\boldsymbol{W}}_{:,j} - ((\boldsymbol{W} - \hat{\boldsymbol{H}})\boldsymbol{\Sigma})_{:,j} \right] / \Sigma_{j,j}$ // $\tilde{\beta}$ from Lemma 1
        for column $j$.  $\boldsymbol{W}$ is substituted with $\boldsymbol{W} - \hat{\boldsymbol{H}}$.
    $\hat{\boldsymbol{W}}\boldsymbol{\Sigma} \leftarrow \hat{\boldsymbol{W}}\boldsymbol{\Sigma} - \hat{\boldsymbol{W}}_{:,j}\boldsymbol{\Sigma}_{j,:}$ // Part $(A)$ of rank-1 update from (6)
    $\hat{\boldsymbol{W}}_{i,j} \leftarrow q_i(-u_i), i \in [q]$ // Perform updates from (4)
    $\hat{\boldsymbol{W}}\boldsymbol{\Sigma} \leftarrow \hat{\boldsymbol{W}}\boldsymbol{\Sigma} + \hat{\boldsymbol{W}}_{:,j}\boldsymbol{\Sigma}_{j,:}$ // Part $(B)$ of rank-1 update from (6)
  **end**
  $\nabla_{\boldsymbol{H}} g(\hat{\boldsymbol{W}}, \hat{\boldsymbol{H}}) \leftarrow 2\hat{\boldsymbol{H}}\boldsymbol{\Sigma} + 2\hat{\boldsymbol{W}}\boldsymbol{\Sigma} - 2\boldsymbol{W}\boldsymbol{\Sigma}$ // Calculate the gradient of $g$
    from (7).
  $\hat{\boldsymbol{H}} \leftarrow P_s(\hat{\boldsymbol{H}} - \eta\nabla_{\boldsymbol{H}} g(\hat{\boldsymbol{W}}, \hat{\boldsymbol{H}}))$ // Perform update (8)
**end**
**return** $\hat{\boldsymbol{W}}, \hat{\boldsymbol{H}}$

---

