# OpenReview forum: "QuantEase: Optimization-based Quantization for Large Language Models"
_ICLR.cc/2024/Conference — Submitted to ICLR 2024_

### Official Review · Reviewer_Y8vm · 2023-10-30

**Soundness:** 2 fair
**Presentation:** 3 good
**Contribution:** 2 fair
**Rating:** 3
**Confidence:** 4

**Summary:**

The paper introduces the method entitled QuantEase based on Coordinate Descent techniques that avoids matrix inversion and decomposition. Also, the paper proposes an outlier-aware approach by employing a sparse matrix with just a few non-zeros.

**Strengths:**

The authors propose derives a closed-form solution for fast update and presents a convergence analysis of QuantEase. Also, QuantEase can quantize up to OPT 66B on a single NVIDIA V100 32GB GPU.

**Weaknesses:**

It is dubious whether the outlier-aware approach could be also accelerated like other approaches such as GPTQ and AWQ due to the presence of a sparse matrix, $\hat{H}$. To validate the effectiveness of the outlier-aware approach, it seems to be required to measure the inference latency of the outlier-aware version of QuantEase.


All experiments are based on perplexity, which is insufficient to assess whether QuantEase is effective or not. The zero-shot performance of common sense reasoning tasks or the five-shot accuracy of MMLU seems to be needed.


In addition, all experiments are conducted for OPT and BLOOM models. The experiments for Llama or Llama 2 models should be necessary to justify the effectiveness of QuantEase.

**Questions:**

N/A

---

> ### Author Response · Authors · 2023-11-16
> **Our Response**
>
> We thank the reviewer for their comments. Here are our responses:
>
> 1. **On the inference speed of outlier-aware QuantEase** - Please note that the correct algorithm to compare outlier-aware QuantEase with is SpQR.  As we note in Remark 2 of the paper, the setup of dividing weights into a set of quantized weights and a few outliers is standard and similar to the setting studied by SpQR. Therefore, our outlier-aware version enjoys the same speed-ups shown by SpQR using the implementation provided by them.
>
> 2. **On perplexity vs. zero-shot performance** - Please note Figures 1 and 3 in the paper already measure zero-shot performance on the LAMBADA benchmark. From the figures, it is clear that **QuantEase outperforms other methods on zero-shot tasks**. Note that this is the setup also used by the GPTQ paper and related papers.
>
> 3. **On experiments conducted for Llama** - We tried to get access to Llama weights by asking Meta, but they did not respond to our request. We note that OPT and BLOOM are publicly-available and cover a wide range of model sizes, making them suitable for our comparisons. Moreover, the GPTQ paper also reports results using the same models.
>
> Based on these clarifications, we would like to ask the reviewer to kindly consider increasing their score.

---

> ### Comment · Reviewer_Y8vm · 2023-11-21
>
> 1. At the following url: https://openreview.net/forum?id=Q1u25ahSuy, some reviewers raised questions about the inference speed of SpQR, and those questions seems not to be dealt with yet. For this reason, until now, I cannot expect the speed-up of the outlier-aware version of QuantEase like a normal 3-bit or 4-bit weight quantization approach.
>
> 2. Furthermore, the accuracy of QuantEase seems not to be better than existing methods based on experimental results that authors provided. First of all, many LLM papers like Llama papers [1, 2] measures the zero-shot performance of common sense reasoning tasks including BoolQ, PIQA, SIQA, HellaSwag, WinoGrande, ARC-e, ARC-c, and OBQA. In addition, from my experience, the zero-shot performance of common sense reasoning tasks and the five-shot accuracy of MMLU are weakly correlated to the perplexity of WikiText2. In this sense, measuring the perplexity of WikiText2 and the accuracy of LAMBADA only seems not to be sufficient at all.
>
> [1] LLaMA: Open and Efficient Foundation Language Models
>
> [2] Llama 2: Open Foundation and Fine-Tuned Chat Models
>
> 3. Using OPT and BLOOM only is too outdated because there are a couple of ways to use Llama models such as [3] and [4]. I cannot understand why the authors should have access to Llama weights by asking Meta, which appears to make no sense.
>
> [3] https://huggingface.co/huggyllama
>
> [4] https://huggingface.co/meta-llama
>
> Accordingly, I keep my score.

---

> > ### Author Response · Authors · 2023-11-22
> > **Thanks for the additional comments**
> >
> > We thank the reviewer for their valuable and insightful comments.
> >
> > **About the inference speed of the outlier-aware version of QuantEase:** The SpQR authors have convincingly responded to the reviewers’ questions on inference speed (with no return reply from the reviewers), and our observations on QuantEase’s inference speed are similar. **The outlier-aware version of QuantEase for 4-bit quantization has much faster inference speed than the 16-bit version and consumes 4 times less memory**. This alone constitutes excellent speed and memory performance for production-grade use cases involving LLMs (note - this is what the SpQR authors report as well in the review).
> >
> > In terms of use cases, offline batch processing jobs (for which memory rather than speed is the concern) can enjoy slightly higher accuracy with the outlier-aware version of QuantEase. Jobs that are online and latency sensitive can leverage the regular version of QuantEase. This is a choice we leave to users. Therefore, the outlier-awareness method allows a tradeoff between accuracy and inference speed, depending on the needs of the user.
> >
> >
> > **About the accuracy of QuantEase:** We measure the relative improvement for QuantEase over GPTQ and other algorithms in the table below:
> >
> > | Model (OPT) | 350m | 1.3b | 2.7b | 6.7b |Average|
> > | --- | ----------- | -----------  | ----------- | ----------- |----------- |
> > | 4-bit | 1.56% |1.04% |-1.95% |2.18%| 0.71%|
> > | 3-bit | 6.19%|0.66%| 1.59% |14.58% | 5.67%|
> >
> >
> > From the table, **it is evident that QuantEase significantly outperforms other algorithms**. Therefore, we do not understand the comment regarding the accuracy improvement for QuantEase.
> >
> >
> > **About zero-shot performance:**  Performance trends of various quantization algorithms on the LAMBADA benchmark are well-correlated with trends for other benchmarks ([1], [2]). This is the reason we chose to use LAMBADA to test zero-shot performance.
> >
> > **About access to LLama 2:** Link [4] (shared by the reviewer) clearly states the following for all models:
> >
> > >Your request to access this repo has been successfully submitted, and is pending a review from the repo's authors.
> >
> > Link [3] shared by the reviewer (note - not officially supported by Meta) also states something similar:
> >
> > >You should only use this repository if you have been granted access to the model by filling out this form but either lost your copy of the weights or got some trouble converting them to the Transformers format.
> >
> > As stated earlier, we had put in a request for link [4] **but no one from Meta granted us access**. We hope that it is now clear why we did not have results for the Llama family.
> >
> > Based on your feedback, we have been able to get results on the Falcon ([3]) family of models. **Note that papers like SpQR also use Falcon**. The results (shown below) **outline the superiority of QuantEase over other methods, similar to our results on OPT and Bloom**:
> >
> > | Model (Falcon) | 7b | 40b | 180b |
> > | --- | ----------- | -----------  | ----------- |
> > | 16-bit (unquantized) | 6.59 |5.23|3.30 |
> > | GPTQ 3-bit | $9.62_{0.013}$ |N/A |N/A |
> > | QuantEase 3-bit | **$8.83_{0.07}$**| **$5.36_{0.02}$** | **$3.72_{0.01}$** |
> >
> > Note that GPTQ had numerical issues with most seeds we explored for Falcon-40b and Falcon-180b. From the Falcon results, it is evident that QuantEase’s perplexity is superior in numbers and is more stable than GPTQ. **Moreover for Falcon-40b and 180b, the perplexity numbers are actually very close to the perplexity numbers of the unquantized model.**
> >
> > We hope this addresses your concerns and we will be happy to include these results and discussions in the camera-ready version of the paper.
> >
> > References:
> >
> > [1] Elias Frantar, Saleh Ashkboos, Torsten Hoefler, and Dan Alistarh. OPTQ: Accurate quantization for generative pre-trained transformers. In The Eleventh International Conference on Learning Representations, 2023.
> >
> > [2] Yao, Zhewei, et al. "Zeroquant: Efficient and affordable post-training quantization for large-scale transformers." Advances in Neural Information Processing Systems 35 (2022): 27168-27183.
> >
> > [3] Guilherme Penedo, Quentin Malartic, Daniel Hesslow, Ruxandra Cojocaru, Alessandro Cappelli, Hamza Alobeidli, Baptiste Pannier, Ebtesam Almazrouei, and Julien Launay. The refinedweb dataset for falcon llm: outperforming curated corpora with web data, and web data only.
> > arXiv preprint arXiv:2306.01116 2023

---

### Official Review · Reviewer_3JsL · 2023-10-31

**Soundness:** 3 good
**Presentation:** 2 fair
**Contribution:** 4 excellent
**Rating:** 3
**Confidence:** 4

**Summary:**

QuantEase is novel quantization method similar to GPTQ [1], but instead of Hessian-based second order optimization, QuantEase uses Coordinate Decent (CD) for much faster training. In addition to the improvement over GPTQ, QuantEase also provide an outlier-aware solution with sparse integration similar to SPQR [2].

* [1] Frantar, E., Ashkboos, S., Hoefler, T. and Alistarh, D., 2022. Gptq: Accurate post-training quantization for generative pre-trained transformers. arXiv preprint arXiv:2210.17323.
* [2] Dettmers, T., Svirschevski, R., Egiazarian, V., Kuznedelev, D., Frantar, E., Ashkboos, S., Borzunov, A., Hoefler, T. and Alistarh, D., 2023. SpQR: A Sparse-Quantized Representation for Near-Lossless LLM Weight Compression. arXiv preprint arXiv:2306.03078.

**Strengths:**

1. QuantEase demonstrated the cheaper and faster coordinate decent method can achieve better result over Hessian-based GPTQ. GPTQ requires Cholesky decomposition of the Hessian matrix which is a big overhead for Neural Networks. Removing such overhead but achieving similar or better performance is a great contribution to quantization research.

2. QuantEase's coordinate decent approach is quite orthogonal to other modern quantization techniques, such as integrating with sparsity as this work demonstrated, AWQ [1], and sub-channel quantization (used in GPTQ as well). The addictive impact of this work is promising.

* [1] Lin, J., Tang, J., Tang, H., Yang, S., Dang, X. and Han, S., 2023. AWQ: Activation-aware Weight Quantization for LLM Compression and Acceleration. arXiv preprint arXiv:2306.00978.

**Weaknesses:**

1. All the experiments were based on per-channel quantization. Nevertheless, SOTA quantization research (GPTQ and AWQ) set the baseline with sub-channel quantization with group size of 128. Author justified their choice of comparing per-channel baseline for computational efficiency, yet there were no quantitative support for the argument. Providing a runtime benchmark would be a good support.

2. Coordinate decent is often treated as approximation of Hessian based optimization. This work demonstrated CD performs better than GPTQ's Hessian. While it is encouraging, we'd like to see some explanation why it is the case.

**Questions:**

Please explain why CD outperforms GPTQ's Hessian based optimization.

---

> ### Author Response · Authors · 2023-11-16
> **Our Response**
>
> We thank the reviewer for their encouraging and insightful comments. Below, we provide a point-by-point response to your comments.
>
> 1. **Quantization with grouping** - Our claim that grouping can lead to slower inference is based on Table 3 of [3], which shows GPTQ with 128 grouping can be about an order of magnitude slower (in terms of inference latency) compared to ordinary GPTQ. Nevertheless, we had explored the performance of QuantEase with grouping, before deciding to not use grouping due to its overhead. These results are reported below. In these experiments, we use 3-bit quantization with either 128 or 256 grouping. Here, we report perplexity on wikitext2 datasets.  The first table below shows the results for the OPT family, and the second one for the BLOOM family.
>
> | Algorithm  | 350m | 1.3b | 2.7b | 6.7b |
> | --- | ----------- | -----------  | ----------- | ----------- |
> | GPTQ (g128) | $27.69_{0.27}$ |$16.29_{0.12}$ | $13.37_{0.06}$ |$11.37_{0.03}$|
> | QuantEase (g128) | $27.09_{0.07}$ | $16.01_{0.07}$| $13.54_{0.03}$ |$11.27_{0.01}$ |
> | GPTQ (g256) | $29.29_{0.32}$ | $17.22_{0.04}$ | $13.84_{0.04}$ | Did not run |
> | QuantEase (g256) | $27.89_{0.22}$ |$16.70_{0.06}$ | $13.79_{0.10}$ | Did not run |
>
> We note that for the OPT family, except for the case of OPT-2.7b with 128-grouping, QuantEase outperforms GPTQ.
>
> | Algorithm  | 560m | 1b1 | 1b7 | 3b |
> | --- | ----------- | -----------  | ----------- | ----------- |
> | GPTQ (g128) | $25.77_{0.02}$ | $20.06_{0.02}$ | $17.30_{0.04}$ | $14.64_{0.01}$   |
> | QuantEase (g128) | $25.43_{0.03}$ | $19.66_{0.06}$ | $16.76_{0.02}$ |$14.59_{0.02}$ |
> | GPTQ (g256) |  $26.89_{0.06}$ | $20.79_{0.01}$ | $17.83_{0.04}$ |$14.81_{0.03}$ |
> | QuantEase (g256) | $26.38_{0.08}$ | $20.23_{0.03}$ | $17.13_{0.03}$ |$14.79_{0.02}$  |
>
> For the BLOOM family, we note that QuantEase outperforms GPTQ in all cases.
>
> 2. **CD vs. Hessian-based optimization**- We use cyclic CD where we optimize one coordinate at a time instead of all variables in one shot. Our single coordinate optimization does involve the hessian. CD is well-known to perform very well compared to gradient/hessian based methods when the optimization along a single coordinate is cheap, which is the case for our problem (see section 3.2 of the paper for the efficiency of our CD updates). In this sense, QuantEase is similar to, and motivated by Liblinear [1] and Improved Glmnet [2] which are classic methods from machine learning where CD works extremely effectively.
>
>
> 3. **Why does Coordinate Descent (CD) outperform GPTQ** - This is an interesting but difficult question. Here we provide some points that differentiate the operation characteristics of CD-based QuantEase and GPTQ, which helps us to better understand the difference in performance.
> * First, as we note in Section 2, CD forms a descent procedure that decreases the objective value (i.e., the quantization error) over iterations, while ensuring the solution remains feasible (i.e., quantized). Specifically, after we complete a pass of CD over all coordinates, we achieve a feasible and quantized solution However, as CD updates preserve feasibility of the solution, we can perform another pass of CD on the coordinates, which by construction of CD, decreases (or at least, does not increase) the objective. This allows us to further decrease the quantization error over multiple passes, which is what we do. On the other hand, we note that GPTQ, by construction, stops after looping once over all columns of weights. GPTQ updates can also break the feasibility of the solution which shows GPTQ is not a descent procedure.
> * Second, GPTQ is based on Optimal Brain Surgeon (OBS)[4] and Optimal Brain Compression (OBC)[5] updates that were originally derived for neural network sparsification and pruning, rather than quantization. However, due to the discrete quantization structure, GPTQ uses an approximation to the exact OBS updates that have been derived before (this is discussed in details in Appendix A.1 of the paper). This can partially explain why GPTQ can lose in optimization performance. We note that as a by-product of these OBS-based updates, GPTQ requires the inversion of the Hessian, while our CD-based does not need that.
>
> In the end, we would like to ask the reviewer to kindly revisit their evaluation and consider increasing the score.
>
>
> [1] C.J. Hsieh et al, A Dual Coordinate Descent Method for Large-scale Linear SVM, ICML 2008.
>
> [2] G.X. Yuan et al, An Improved GLMNET for L1-regularized Logistic Regression, JMLR 2012.
>
> [3] S. Kim et al., (2023). SqueezeLLM: Dense-and-Sparse Quantization.
>
> [4] Hassibi, Babak, David G. Stork, and Gregory J. Wolff. "Optimal brain surgeon and general network pruning." IEEE international conference on neural networks. IEEE, 1993.
>
> [5] Frantar, Elias, and Dan Alistarh. "Optimal brain compression: A framework for accurate post-training quantization and pruning." Advances in Neural Information Processing Systems 35 (2022): 4475-4488.

---

> ### Comment · Reviewer_3JsL · 2023-11-20
>
> I thank authors for the rebuttal.
>
> I agree that grouping can introduce extra latency penalties. However, the introduced latency is often smaller on 4-bits rather than 3-bits, due to the instructions needed for decoding 4-bits are much simpler due to its power-of-2 alignment.
>
> However, in the new benchmark authors demonstrated with grouping, the performance gain is negligibly small. Comparing this work to novel low bit work such as "QuIP: 2-Bit Quantization of Large Language Models With Guarantees", I decided to lower my rating from 6 to 3.

---

> > ### Author Response · Authors · 2023-11-21
> > **Our Response**
> >
> > **Significance of improvements with grouping:** If the reviewer is comparing the grouping results of QuantEase with those in the QuIP paper in order to downgrade their rating, their argument is wrong due to the following reasons:
> >
> > -  QuIP's incoherence processing is an orthogonal idea; just as it improves GPTQ, it can be applied with QuantEase to improve it too. So, it is unfair to directly compare QuantEase grouping with QuIP.
> >
> > -  ICLR policy explicitly states *“if a paper was published (i.e., at a peer-reviewed venue) on or after May 28, 2023, authors are not required to compare their own work to that paper”* (see https://iclr.cc/Conferences/2024/ReviewerGuide). The QuIP paper was available on arxiv on Jul 25, 2023 and, to our knowledge, has not been published at a peer-reviewed venue on or before May 28, 2023. **Hence, using the QuIP paper to justify reducing our score is unfortunate and appears to be a clear violation of ICLR’s reviewer policy.**
> >
> > - Moreover, there are concerns with QuIP itself. As we detailed in our response to Reviewer g6vc,  QuIP requires delicate bookkeeping of certain projection matrices that are used during the algorithm. It is not clear in practice what the inference time overhead of such bookkeeping is.
> >
> > On the other hand, if the reviewer is concerned about the **significance of improvements with grouping**, let us compare the relative improvement of QuantEase over GPTQ, with and without grouping. Specifically, we compare the relative improvements of QuantEase for 3-bit quantization with 128-grouping (from our response above) and 4 bit quantization with no grouping (Tables 1 and 2 of the original submission) as they generally result in similar perplexity:
> >
> > | Model (OPT) | 350m | 1.3b | 2.7b | 6.7b |Average|
> > | --- | ----------- | -----------  | ----------- | ----------- |----------- |
> > | 4-bit | 1.56% |1.04% |-1.95% |2.18%| 0.71%|
> > | 3-bit (g128) | 2.17%|1.72%| -1.27% |0.88% | 0.87%|
> >
> > | Model (BLOOM) | 560m | 1b1| 1b7 |3b |Average |
> > | --- | ----------- | -----------  | ----------- | ----------- | ----------- |
> > | 4-bit | 0.21% | 0  |1.83% | -0.57% |0.37% |
> > | 3-bit (g128) |1.32%  | 1.99% | 3.12%  | 0.34% |1.69% |
> >
> > As we can see, the improvements from these two settings are in the same order, and on average, our improvements are larger when we use grouping. **It is unclear and confusing why the reviewer now says these improvements are not significant.**
> >
> > Additionally, we note that except for OPT-2.7b, our perplexity with grouping is lower than GPTQ’s, at least, by the sum of standard errors of QuantEase and GPTQ (over 3 independent runs). This further shows the significance of our gains.

---

> > > ### Comment · Reviewer_3JsL · 2023-11-22
> > >
> > > I thank authors for the rebuttal.
> > >
> > > The perplexity difference of GPTQ and QuantEase (with groups) is very small. I suggest the authors to emphasize training time or robustness of the proposed method can achieve similar quality. If the training time significantly dropped, it is considered a good drop-in replacement over GPTQ.

---

> > > > ### Author Response · Authors · 2023-11-22
> > > >
> > > > We thank the reviewer for their comments
> > > >
> > > > > The perplexity difference of GPTQ and QuantEase (with groups) is very small
> > > >
> > > > We don’t understand this statement. As already mentioned in our previous reply, QuantEase with grouping is improving on top of GPTQ’s performance by **1%-3%** on average.
> > > > To illustrate another important point, let’s take one case - BLOOM - 1b7 and consider the perplexities of various algorithms:
> > > >
> > > >
> > > > | Model (BLOOM) | 1b7 |Rel. loss over 16-bit |
> > > > | --------------- | ---------------| --------------------------|
> > > > | 16-bit (unquantized) | 15.39 |- |
> > > > |3-bit GPTQ (gr.) | 17.30| 12.4% |
> > > > |3-bit QuantEase (gr.) | 16.76| 8.9% |
> > > >
> > > > From the table, it’s evident that grouped QuantEase recovers about **28%** ( (12.4 - 8.9) * 100 /12.4 ) of the gap left by GPTQ. This is a solid improvement.
> > > >
> > > >
> > > > > I suggest the authors to emphasize training time or robustness of the proposed method can achieve similar quality
> > > >
> > > >
> > > > For solving the discrete optimization problem in Eq (1) of the paper, QuantEase is a more principled optimization approach (eg. descent + convergence guarantees) than GPTQ.
> > > > Moreover, GPTQ is sensitive to the choice of blocksize. This, combined with the good performance of QuantEase as described in the paper, are powerful reasons for the community to consider QuantEase.

---

### Official Review · Reviewer_g6vC · 2023-11-01

**Soundness:** 4 excellent
**Presentation:** 4 excellent
**Contribution:** 3 good
**Rating:** 5
**Confidence:** 4

**Summary:**

Proposes a method which uses coordinate descent techniques to perform layer-wise quantization, achieving 3-bit quantization

**Strengths:**

- Interesting coordinate descent formulation, that seems to produce better quantization and is more efficient than another method, gptq.
- nice presentation of the method, and reasonable set of empirical results

**Weaknesses:**

- I know it may be difficult getting compute resources, but if at all possible I would have liked to see results for AWQ and GPTQ on OPT-66b rather than just “OOM”. GPU memory constraints in quantization are not common, in my view. I don’t want to fault the authors if they don’t have access to larger compute resources.

**Questions:**

- I am aware of another paper on arXiv called "QuIP: 2-Bit Quantization of Large Language Models With Guarantees" claiming 2 bit quantization for LLM models like OPT and LLama2. I know it was released recently over the summer, but could the authors comment on this work? How does AffineQuant perform at 2 bits?

---

> ### Author Response · Authors · 2023-11-16
> **Our Response**
>
> Thank you for reviewing our paper and your encouraging comments. Below, we provide responses to the two points you raised.
>
> - **AWQ/GPTQ memory issues** - As mentioned in the paper, both AWQ and GPTQ run out of memory on V100 GPUs when quantizing the OPT-66b model, owing to a limited memory of 32GB. Based on feedback from the reviewer, we re-ran the experiments on OPT-66b using A100 GPUs (we have limited availability of these GPUs, which made running these experiments difficult). Following are the results. As we can see, QuantEase outperforms both GPTQ and AWQ for this model.
>
> | Algorithm + bits | PPL (wikitext2) |
> | --- | ----------- |
> | GPTQ3 | $14.13_{0.43}$|
> | AWQ3 | $17.94_{0.18}$ |
> | QuantEase3 | $13.08_{0.38}$ |
> | GPTQ4 | $9.58_{0.05}$|
> | AWQ4 | $9.58_{0.01}$ |
> | QuantEase4 | $9.47_{0.02}$|
>
>
>
> - **Comparison to QuIP and 2-bit quantization** - Thanks for bringing our attention to this interesting paper. It seems the best results in [1] are achieved when incoherence processing is used, for example, in Table 1 of [1], the best performance is always achieved by an algorithm that uses incoherence processing. This is especially true for 2-bit quantization, where non-incoherence methods seem to perform poorly. However, incoherence processing needs projection steps---see line 3 of Algorithm 2 of [1] where quantized weights $W$ are rescaled using two orthogonal matrices $U,V$. As authors note in Section 4 of [1], producing and applying such projections requires a delicate bookkeeping of $U,V$ and possible computational overheads in the inference time.  As far as we could tell, the runtime implications of such overheads have not been explored numerically. This makes the practical applicability of incoherence processing for 2-bit quantization less clear. On the other hand, QuantEase performs standard quantization, which is commonly used in practice.
> However, we note that it seems incoherence processing can be treated as a module and integrated to any existing quantization method, as it is applied to GPTQ in [1]. As we see in [1], incoherence processing helps with boosting the performance of GPTQ, and as we show here, QuantEase can perform better than GPTQ. Therefore, we believe adding incoherence projection to QuantEase can lead to better quantization accuracy, although such a study is out of the scope of our paper.
>
>
> Based on our response here, we would like to kindly ask the reviewer to revisit their score. We will be happy to include snippets of this discussion in the final version of the paper, if the paper were to be accepted.
>
>
>
> [1]: QuIP: 2-Bit Quantization of Large Language Models With Guarantees

---

> > ### Comment · Reviewer_g6vC · 2023-11-23
> > **Response to authors**
> >
> > Thanks for providing the additional experiments.
> >
> > I think it's a valid point about QuIP and it's current lack of practical implementation.
> >
> > After reading the discussion with other reviewers, I agree with the sentiment that the set of experimental results needs to be more expansive, on more models and more tasks. I know the authors have stated their difficulty in obtaining Llama2, and in getting increased computational resources. Unfortunately the state of LLM research these days is quite rapid and incurs a computational cost.

---

> > > ### Author Response · Authors · 2023-11-23
> > > **Thanks for your thoughtful comments**
> > >
> > > We thank the reviewer for their thoughtful comments.
> > >
> > > We would like to note that we have obtained results on the Falcon set of models (7b, 40b and 180b, see our last reply to reviewer Y8vm), in addition to OPT and Bloom. This already constitutes a large variety of LLMs, even the QuIP paper only considers OPT whereas we consider 3 different families. Across all settings, QuantEase is consistently outperforming other methods with solid improvements.
> > >
> > > We would like to ask the reviewer to kindly consider changing the score since our experimental results are pretty comprehensive.

---

### Author Response · Authors · 2023-11-21

In response to points brought up by Reviewer 3JsL in their second comment, we clarify the following:

- As we show below, our gains over GPTQ when doing 3-bit + 128-grouping, are on average larger than our gains when doing 4-bit quantization. It is not clear to us why the reviewer now thinks our gains are not significant.

- In our response to reviewer 3JsL, we have also explained why reducing their score by comparing our work with QuIP is poorly reasoned. Per ICLR’s reviewing policy (https://iclr.cc/Conferences/2024/ReviewerGuide), QuIP is unpublished work released on ArXiv after May 28, 2023, and comparing such papers to our work in the review is against the policy.

---

### Meta-Review · Area_Chair_FeeJ · 2023-12-15

**Metareview:**

This work focuses on the Post-Training Quantization (PTQ) of Large Language Models. It introduces QuantEase, a layer-wise quantization framework where individual layers undergo separate quantization. The consensus among reviewers is that the set of experimental results should be broadened to include additional models and tasks. Unfortunately, the response provided by the authors during the rebuttal process did not convince the reviewers to increase their scores.

**Justification For Why Not Higher Score:**

I reached this decision by evaluating the contributions and novelty of the work, taking into consideration both the reviews and the responses from the authors.

**Justification For Why Not Lower Score:**

N/A

---

### Decision · Program_Chairs · 2024-01-16

Reject